# Uncovering the impact of hyperparameters for global magnitude pruning

## Abstract

A common paradigm in model pruning is to train a model, prune, and then either fine-tune or, in the lottery ticket framework, reinitialize and retrain. Prior work has implicitly assumed that the best training configuration for model evaluation is also the best configuration for mask discovery. However, what if a training configuration which yields *worse* performance actually yields a mask which trains to *higher* performance? To test this, we decoupled the hyperparameters for mask discovery ($H_{find}$) and mask evaluation ($H_{eval}$). Using unstructured magnitude pruning on vision classification tasks, we discovered the "decoupled find-eval phenomenon," in which certain $H_{find}$ values lead to models which have lower performance, but generate masks with substantially higher eventual performance compared to using the same hyperparameters for both stages. We show that this phenomenon holds across a number of models, datasets, configurations, and also for one-shot structured pruning. Finally, we demonstrate that different $H_{find}$ values yield masks with materially different layerwise pruning ratios and that the decoupled find-eval phenomenon is causally mediated by these ratios. Our results demonstrate the practical utility of decoupling hyperparameters and provide clear insights into the mechanisms underlying this counterintuitive effect.

## 1 Introduction

There has been significant progress in deep learning in recent years, but many of the best performing networks are extremely large (Kolesnikov et al., 2019; Brown et al., 2020). This can be problematic due to the amount of compute and memory needed to train and deploy such models. One popular approach is model pruning: removing weights (unstructured pruning) or units (structured pruning) from a trained network in order to generate a smaller network with near-equivalent (and in some cases, better) performance (Blalock et al., 2020; Lin et al., 2020; Liu et al., 2017; He et al., 2019b; Molchanov et al., 2019). One of the most commonly used heuristics is magnitude pruning, in which the lowest magnitude weights/units are removed, as it is simple and competitive with more complex methods (Han et al., 2015; Gale et al., 2019).

The lottery ticket hypothesis (Frankle & Carbin, 2018), a related concept, posits that a large network contains a smaller subnetwork at initialization that can be trained to high performance in isolation, and provides a simple pruning method to find such winning lottery tickets (LTs). In addition to allowing the training of sparse models from scratch, the existence of LTs also suggests that overparameterization is not necessarily required to train a network to high performance; rather, overparameterization may simply be necessary to find a good starting point for training.

When pruning a network, one must decide how much to prune each layer. Should all layers be pruned equally? Or rather, should some layers be pruned more than others? Previous studies have shown that global pruning results in better compression and performance than layerwise (or uniform) pruning (Frankle & Carbin, 2018; Morcos et al., 2019). This is because global pruning ranks all weights/units together independent of layer, granting the network the flexibility to find the ideal layerwise pruning ratios (LPR), which we define as the percent that each layer is pruned.

One can frame most pruning methods as having two phases of training: one to find the mask, and one to evaluate the mask by training the pruned model (whether by rewinding and re-training or fine-tuning). Common practice for methods that rewind weights after pruning, such as lottery ticket pruning, has been to use the same hyperparameters for both finding and evaluating masks (see Blalock

et al. (2020) for an extensive review). Methods that fine-tune weights after pruning typically train at a smaller learning rate than the training phase to find the mask (Han et al., 2015; Renda et al., 2020), but other hyperparameters are held constant.

Pruning methods also rest on the assumption that models with the best performance will generate the best masks, such that the optimal hyperparameters for mask generation and mask evaluation should be identical. However, what if the mechanisms underlying mask generation are not perfectly correlated with those leading to good performance? Consequentially, what if models which converge to *worse performance* can yield *better masks*? To test this, we explored settings for global magnitude pruning in which different hyperparameters were used for mask generation and mask evaluation. In particular, we focused on three hyperparameters that are commonly adjusted in practice: learning rate, batch size, and weight decay. Using this paradigm, we make the following contributions:

- Surprisingly, we found that the best hyperparameters to find the mask ($H_{find}$) are often different from the best hyperparameters to train the regular model or to evaluate the mask ($H_{eval}$; Figures 1 and 2), which we term the "decoupled find-eval phenomenon". Counter-intuitively, this means that models with worse performance prior to pruning can generate masks which leads to better performance for the final pruned model than a mask generated by a higher performance model pre-pruning.

- We show that this phenomenon is not an artifact of the particular setting we studied and also occurs for structured pruning (Figure 5), other datasets, and other variants of LTs, including late rewinding, learning rate warmup, and learning rate rewinding (Figure 4).

- We found that different hyperparameters for mask generation led to materially different LPRs (Figures 6a and A18). Notably, we observed that a larger learning rate, smaller batch size, and larger weight decay (which resulted in worse masks) consistently pruned early layers far more than in masks found with the opposite.

- Finally, we show that this phenomenon is causally mediated by the differences in LPR. When the LPR is fixed to a "good" LPR (i.e., that of a high performance mask), the previously bad hyperparameters now lead to better performance *and* better mask generation (Figure 7). The same is true for the inverse experiment.

## 2 METHODS

### 2.1 MODIFIED PRUNING PROCEDURES

Our main experiments are mainly based on the lottery ticket procedure from Frankle & Carbin (2018) along with follow up work (Renda et al., 2020) for unstructured pruning. We used global iterative magnitude pruning (IMP) for LTs because it has been shown to perform better than one shot pruning, where the pruning procedure is only done once rather than iteratively, and local (or uniform layerwise) pruning, where each layer has the same pruning ratio (Morcos et al., 2019; Frankle & Carbin, 2018). We also investigated structured one-shot pruning, following Liu et al. (2017), which also prunes globally. All experiments use magnitude pruning.

We define four different sets of hyperparameters: $H_{unpruned}$, which is used for regular training of an unpruned model; $H_{find}$, used to find masks (i.e. the pre-training part of the pruning procedure); $H_{eval}$, used for obtaining the final pruned model to be used for inference and evaluating the final performance of a mask; and $H_{LT}$ to refer to the hyperparameters optimized for LTs where $H_{find} \triangleq H_{eval}$. Note that $H_{LT}$ is often different from $H_{unpruned}$ in practice and in previous literature (Liu et al., 2019; Frankle & Carbin, 2018). Our modified procedures can incorporate changes in any hyperparameters, but in this paper we focus on experiments with learning rate (LR), batch size (BS), and weight decay (WD). For the main experiments, we focus on one hyperparameter at a time. When only one hyperparameter is changed from $H_{find}$ and $H_{eval}$, we will denote it such as $LR_{find}$, $WD_{eval}$, etc.

**Unstructured pruning** To account for these distinct sets of hyperparameters, we slightly modified the procedure for unstructured IMP as described in Algorithm A1. We emphasize that our modification requires no additional compute compared to the original LT procedure if used practically. However, to generate additional data points for analysis in the present work, we separately evaluated

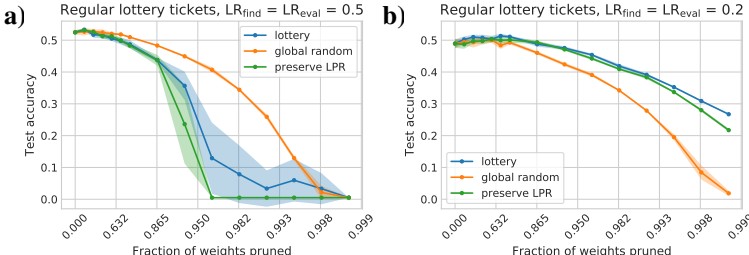

**Figure 1: Test accuracy at various sparsity levels, with the regular lottery ticket procedure, i.e. $LR_{find}$ = $LR_{eval}$.** "Lottery" is the selected mask (winning ticket), "global random" is a mask selected at random globally, and "preserve LPR" is a random mask that has the same LPR as the winning ticket. (**a**) LR = 0.5 performed very poorly, even worse than the random baseline. (**b**) Curves we would expect from a properly functioning lottery ticket. The pruned performance is much better, even though training a regular model without pruning is better with LR as 0.5 than 0.2.

many of the intermediate masks as well. We pruned weights with the smallest magnitudes throughout the whole network (i.e., global magnitude pruning) by $p = 20\%$ at each pruning iteration until we reached 99.9% pruned, for a total of 30 pruning iterations. For some experiments, we also modified line 6 for techniques such as late rewinding and learning rate rewinding (Renda et al., 2020). The original LT algorithm is recovered in Algorithm A1 when $H_{eval} = H_{find}$.

**Structured pruning** For structured pruning, we consider one-shot pruning using the scale of batch normalization layers as described in Algorithm A2. This also has no additional compute requirement compared to the original, and the original is again recovered when $H_{eval} = H_{find}$.

## 2.2 MODELS AND DATASETS

For our main results, we used ResNet-50 (He et al., 2016) on Tiny ImageNet[1] (Le & Yang, 2015), and we also ran experiments with ResNet-18 on Tiny Imagenet and ResNet-50 on MiniPlaces (Zhou et al., 2017). We only pruned convolutional layers, such that the fully connected output layer of ResNets remained unpruned. Unless otherwise stated, we used SGD with 0.9 momentum and trained for 110 epochs, decaying the learning rate with a gamma of 0.2 at epochs 70, 85, and 100. See Section A.1 for details. The defaults for $H_{find}$ and $H_{eval}$ are batch size of 512, learning rate of 0.2, and weight decay of 1e-4; adjustments will be explicitly noted.

## 2.3 RANDOM BASELINES

We compare the selected masks to two random baselines. The first, "global random", selects a mask at random, uniformly throughout the whole network. This means that all layers have the same expected pruning ratio. The second, "preserve LPR", also selects a mask at random, but with the same LPR as the winning ticket that we compare to; this is equivalent to shuffling the winning ticket mask within each layer. Both baselines are evaluated with new randomly initialized weights. Shaded error bars throughout represent standard deviation across 5 runs.

## 3 RESULTS

### 3.1 FAILING LOTTERY TICKETS

The failure of large LRs for lottery tickets has been pointed out previously (Liu et al., 2019); winning tickets were only better than random baselines at a small LR. Directionally consistent with this work, we found a particularly extreme case: not only can random baselines match the performance of the winning tickets with large LR, they can be significantly better. We found these catastrophically failing lottery tickets by following the LT procedure using a learning rate of 0.5 (for both $LR_{find}$ and $LR_{eval}$), as in $H_{unpruned}$. This is illustrated in Figure 1a, where the winning tickets (blue) and the random baseline with the same LPR as the winning ticket (green) are both worse than the

---

[1]Use of these datasets is only for noncommercial, research purposes, and not for training networks deployed in production or for other commercial uses

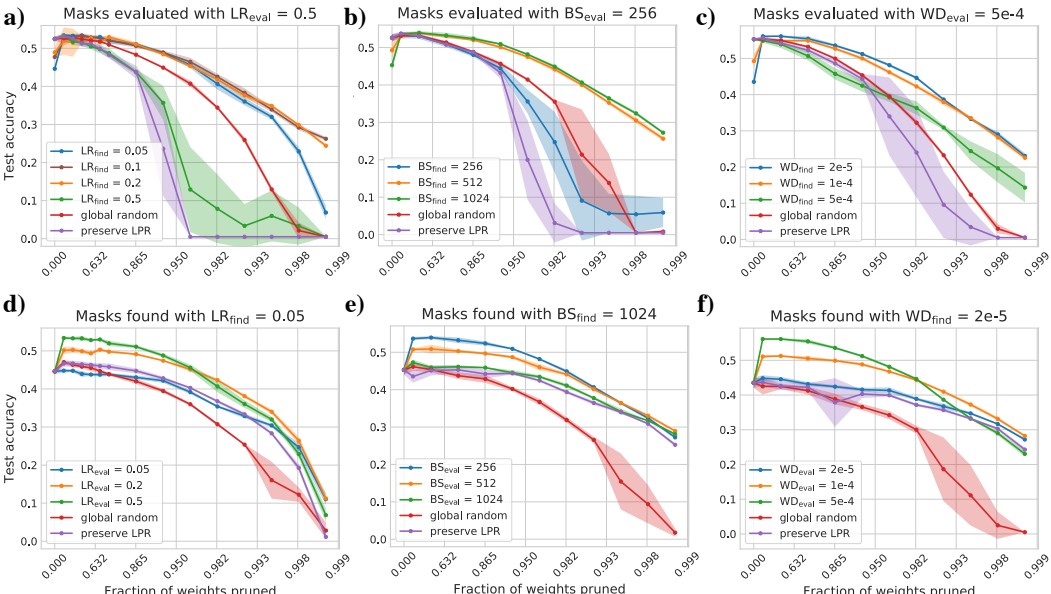

**Figure 2:** Test accuracy for different combinations of $H_{find}$ and $H_{eval}$, adjusting learning rate (**a, d**), batch size (**b, e**), or weight decay (**c, f**). Top row (**a-c**): given a constant $H_{eval}$, we compare different values of $H_{find}$. Bottom row (**d-f**): given a constant $H_{find}$, we compare different values of $H_{eval}$. Random baselines have the same $H_{find}$ and $H_{eval}$ as the value in the title. The points at 0% sparsity represent the unpruned model trained with the same $H_{find}$ as the rest of each line. Overall, it is bad for $H_{find}$ to have a large LR, small BS, or large WD, but the opposite is true for $H_{eval}$. See Figures A2-A6 for additional combinations.

global random baseline (orange) by over 25 percentage points at 96-98% sparsity. Surprisingly, this winning ticket is much worse than the global random baseline, rather than just having no significant improvement over the random baseline as found in Liu et al. (2019) and Frankle & Carbin (2018). The learning rate of 0.5 is not unreasonably large – in fact, the unpruned model performs the best at LR = 0.5. This failure also arises for reasonable values of batch size and weight decay (Figure A1).

Prior works have addressed this problem by training LTs with smaller learning rates ($H_{LT}$ instead of $H_{unpruned}$, where for both, $H_{find} \triangleq H_{eval}$). Similarly in our experiments, dropping the LR from 0.5 to 0.2 resulted in winning tickets that substantially outperform the global random baseline and are somewhat better than the random baseline with the same LPR as the winning ticket (Figure 1b). This is why, in the lottery ticket setting, $H_{LT}$ is typically different from $H_{unpruned}$ (Frankle & Carbin, 2018). However, training the standard unpruned model with a LR of 0.2 instead of 0.5 decreases performance from 52.5% to 48.9%, as depicted by comparing the leftmost points in Figure 1. The accuracy is decreased for low sparsity pruned networks as well. Current practice therefore faces a difficult trade-off between low sparsity and high sparsity model performance.

These findings raise questions in two directions. First, for practical purposes, can we get the best of both worlds by decoupling the hyperparameters used to find the mask ($H_{find}$) and to evaluate the mask ($H_{eval}$)? This is inspired by the fact that large LR/small BS/large WD does well at low sparsity levels but causes the pruned model to get worse as sparsity increases, suggesting that there may be potential for better winning tickets. Second, from a scientific standpoint, how do hyperparameters affect pruning performance, and why does $H_{unpruned}$ not work well for lottery tickets?

### 3.2 THE DECOUPLED FIND-EVAL PHENOMENON

To test the whether we can adjust hyperparameters to improve the performance of winning tickets, we decoupled $H_{find}$ and $H_{eval}$ by setting learning rate (LR), batch size (BS) and weight decay (WD) independently for mask generation and evaluation. Specifically, we evaluated $LR_{find}$, $LR_{eval}$ $\in \{0.05, 0.2, 0.5\}$; $BS_{find}$, $BS_{eval} \in \{256, 512, 1024\}$; and $WD_{find}$, $WD_{eval} \in \{2e\text{-}5, 1e\text{-}4, 5e\text{-}4\}$ for a total of nine find-eval combinations for each hyperparameter.

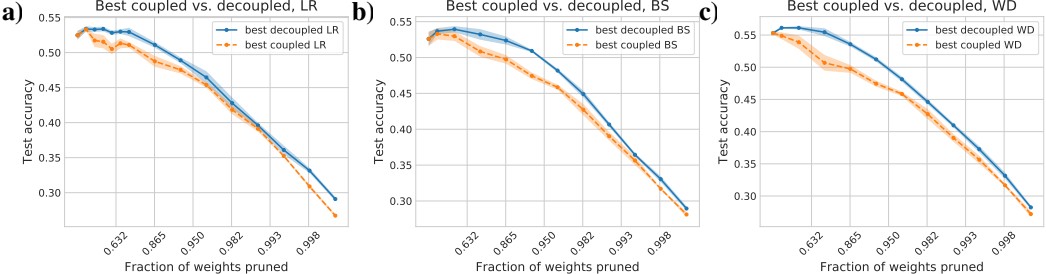

**Figure 3:** Comparison of the best coupled vs. decoupled run, for all three hyperparameters. For clarity, each point at a given sparsity level is mean and standard deviation of the best configuration of any decoupled (blue) or coupled (orange) combination of $H_{find}$, $H_{eval}$ at that sparsity level. This was done because the best $H_{eval}$ changes with sparsity level, and this was cleaner than plotting several lines.

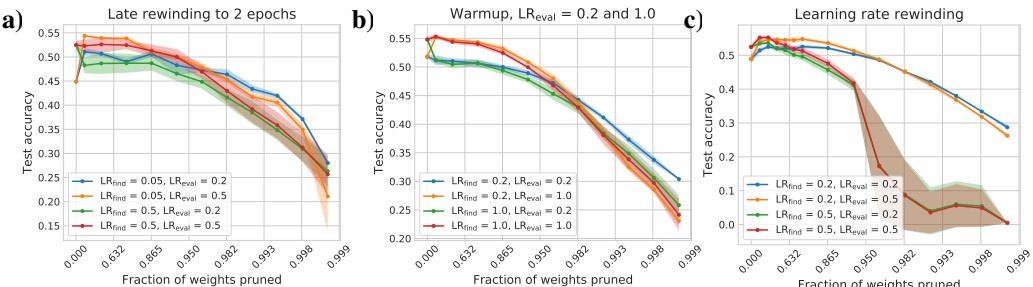

**Figure 4: Test accuracy on variants of LT, with two values of $LR_{find}$ and two values of $LR_{eval}$ each.** (**a**) Late rewinding to 2 epochs instead of initialization. (**b**) Learning rate warmup. (**c**) Learning rate rewinding, where weights continue training from their values during pruning rather than rewinding back to an earlier value. The better $LR_{find}$ for pruned models is the smaller one (blue is better than green, orange is better than red), despite performing worse on the unpruned model, and the best $LR_{eval}$ starts high and then decreases.

The results are shown in Figure 2 grouped from two different perspectives. In the top row (a-c), we compare different values of $H_{find}$ for a constant $H_{eval}$, with one plot for each hyperparameter. These comparisons show that it is better to find masks with a smaller $LR_{find}$ (0.2 is slightly better than 0.05, but both are significantly better than 0.5), larger $BS_{find}$, and smaller $WD_{find}$. For ease of discussion, we will denote these as "good $H_{find}$." In the bottom row (d-f), we compare different values of $H_{eval}$ for a constant $H_{find}$. Here, we observe the opposite effect: given a constant $H_{find}$, the best performance is obtained by a larger $LR_{eval}$, smaller $BS_{eval}$, and larger $WD_{eval}$; we will denote these as "good $H_{eval}$". More specifically, the best $H_{eval}$ matches $H_{unpruned}$ for lower sparsity levels but gradually shift to more intermediate values at higher sparsity levels (around 96-98%).

It is interesting to note that decreasing the batch size has a similar effect as increasing the learning rate, which supports previous studies about the relationship between the two (Smith et al., 2018; Goyal et al., 2017; He et al., 2019a) Increasing weight decay also has a similar effect as increasing the learning rate, corroborating with Zhang et al. (2019). Altogether, these results demonstrate two key findings: first, the good $H_{find}$ values do not achieve the highest model accuracy, which means that hyperparameter tuning on a regular unpruned models ($H_{unpruned}$) may not be relevant when carried over to pruning. Second, the best $H_{find}$ values are not the same as the best $H_{eval}$ values, thus it is important to decouple the $H_{find}$ and $H_{eval}$ in LT pruning. The importance of decoupling can be seen clearly when we compare the best decoupled configuration to the best coupled (original LT) runs (Figure 3): decoupled is better by about 3-4 percentage points around sparsity levels 70-90%.

### 3.3 THE DECOUPLED LR PHENOMENON IS MAINTAINED ACROSS DIFFERENT MODELS, DATASETS, CONFIGURATIONS, AND STRUCTURED PRUNING

**Other models and datasets** To expand beyond ResNet-50 on Tiny ImageNet, we also evaluated a ResNet-50 on MiniPlaces (Figure A8) and a ResNet-18 on Tiny ImageNet (Figure A9). In both cases, we observed the same effect: the best $H_{find}$ requires smaller LR, larger BS, and smaller WD compared to $H_{unpruned}$ and $H_{eval}$.

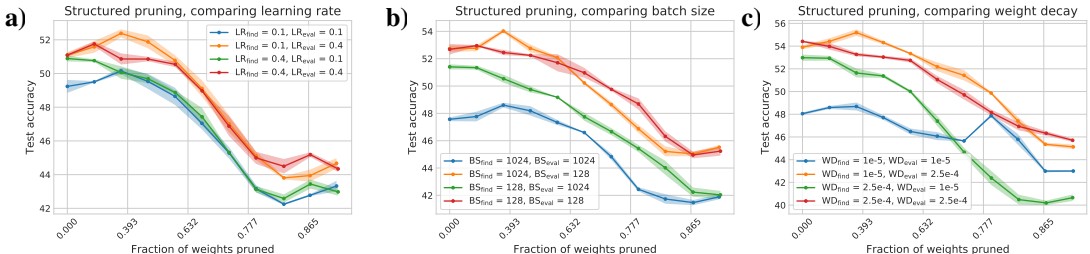

**Figure 5: $H_{find}$ should also be decoupled from $H_{eval}$ in one-shot structured pruning.** Test accuracy for two values of $H_{find}$ and two $H_{eval}$ for four combinations each for learning rate (**a**), batch size (**b**), and weight decay (**c**). The best performing configuration requires decoupled values; though this only applies to low sparsity levels for batch size, the pattern is clearer for learning rate and weight decay. See Figures A11-A12 for full results.

**Other variants of lottery ticket** For the same model and dataset as our initial results (ResNet-50 on Tiny ImageNet), we tried the following: using late rewinding, learning rate rewinding, and learning rate warmup. We ran late rewinding experiments from Renda et al. (2020), where instead of rewinding the weights to $\theta_0$ in line 6 of Algorithm A1, we set the weights to their values at epoch 2. Following Frankle et al. (2019b), we also found that training from this point on was stable, such that we could linearly between modes without substantial drops in performance, though notably this was only true for small learning rates (Figures A13, A14). Epoch 2 was chosen to be roughly similar to experiments in Morcos et al. (2019), which showed strong winning ticket performance across a number of datasets. Here, we again see the same pattern, where the best $LR_{find}$ is smaller than the best standard LR (Figure 4a). For learning rate warmup, we warmed up the learning rate linearly over 1000 iterations from 0 to the specified $LR_{find}$. The effect of $LR_{find}$ is the same as well, although it is more subtle (Figure 4b).

Our learning rate rewinding experiments also encapsulate the fine-tuning experiments (Renda et al., 2020). That is because both methods prune and then continue training the remaining weights from that point; the only difference is that learning rate rewinding trains at a large learning rate, and fine-tuning trains at a small learning rate (due to the decay in learning rate in the training phase before pruning). Thus, we extended learning rate rewinding to different LRs. We saw the same effect, with even better overall performance: taking a mask from a small $LR_{find}$ and using learning rate rewinding to a high $LR_{eval}$ improves accuracy by approximately 3 percentage points compared to weight rewinding (Figure 4c). This also explains why fine-tuning in Renda et al. (2020) did worse than learning rate rewinding: a larger $LR_{eval}$ is better, and fine-tuning is effectively the same as learning rate rewinding but with a smaller and worse $LR_{eval}$.

**Structured pruning** We have demonstrated that our results in Section 3.2 are not merely an artifact of the precise setting we studied, but rather represent a more generic feature of lottery tickets across models, datasets, and configurations. However, this phenomenon may still be limited to only lottery tickets. To test its applicability to pruning more generally, we also investigated structured pruning on ResNet-18 Tiny ImageNet in which entire filters are removed. Once again, we found that the best performing pruned networks required different $H_{find}$ and $H_{eval}$ (Figure 5), though the improvement is more subtle. Like before, we found that it is generally better to find the mask with a smaller $LR_{find}$ and $WD_{find}$ and larger $BS_{find}$, and evaluate with a larger $LR_{eval}$ and $WD_{eval}$ and smaller $BS_{eval}$. The decoupled configuration can increase the top-1 accuracy by up to 2 percentage points at moderate sparsity levels. Coupled hyperparameters were often best at high sparsity levels, but at that point, performance across all configurations were subtantially lower than the unpruned model. These results demonstrate that the phenomenon of decoupling hyperparameters does not just apply to lottery tickets, but also applies to additional methods of global magnitude pruning.

### 3.4 LAYERWISE PRUNING RATIOS CAUSALLY MEDIATE THE DECOUPLED LR PHENOMENON

In Figure 1a, we observed that using a high LR for $H_{LT}$ resulted in performance below the global random baseline not only for the lottery ticket, but also for the "preserve LPR" random baseline. This suggests that LPR may be related to the decoupled find-eval phenomenon. To test this, we examined the LPR over different pruning iterations for various settings. Consistent with our hypothesis, we found that LPR varied dramatically across different $H_{find}$ (Figure 6). Most strikingly, we observed

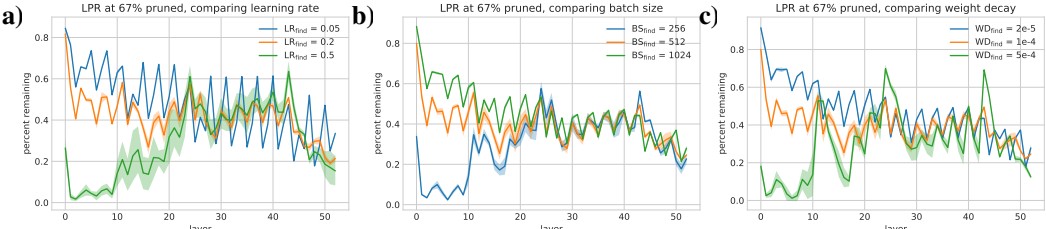

**Figure 6: Layerwise pruning ratios depend on H_find.** Shown here is the LPR of masks found by the three different values of $LR_{find}$ (**a**), $BS_{find}$ (**b**), and $WD_{find}$ (**c**). Small LR, large BS, and small WD values tend to prune more in the earlier layers. See Figure A17 in the appendix for equivalent plots at other sparsity levels.

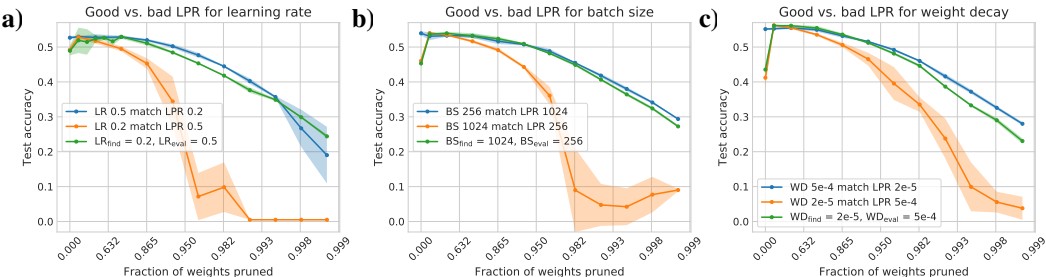

**Figure 7: Test accuracy when LPR is constrained.** For each of learning rate (**a**), batch size (**b**), and weight decay (**c**), the blue line represents pruning using bad $H_{find}$ but LPR fixed to a run from good $H_{find}$. The orange line represents pruning using good $H_{find}$ but LPR fixed to a run from bad $H_{find}$, which performs much more poorly. For comparison, the green lines are regular pruning runs (from Figure 2) that have the same LPR as the blue line, except with a different $H_{find}$ used for finding the masks within each layer. All three lines in each plot are evaluated with the same $H_{eval}$.

that the bad $H_{find}$ values (large $LR_{find}$, small $BS_{find}$, and large $WD_{find}$) tend to prune the earlier layers much more. This imbalance was consistent across different sparsity levels (Figure A17).

We have shown there is a clear difference in LPR across various $H_{find}$, but is the change in LPR directly responsible for the performance of the pruned models? If this is the case, we should be able to change the performance of models just by constraining the model's LPR. To test this, we ran two types of experiments. Based on the results of Figure 2, we consider the LPR of masks from $LR_{find}$ = 0.2, $BS_{find}$ = 1024, and $WD_{find}$ = 2e-5 as "good LPR". Likewise, the LPR of masks from $LR_{find}$ = 0.5, $BS_{find}$ = 256, and $WD_{find}$ = 5e-4 can be considered as "bad LPR". We ran Algorithm A1 with bad $H_{find}$ but forced the pruning step to prune a fixed amount each layer, specifically, according to the good LPR from the runs with respective good $H_{find}$ ("match-good-LPR"). While the LPR are fixed, the masks within each layer were still determined by the particular training configuration (the bad $H_{find}$ in this case). We also ran the complementary procedure with good $H_{find}$ values but pruning according to the bad LPR ("match-bad-LPR").

If LPR causally mediates the decoupled LR phenomenon, we would expect to see match-good-LPR do well and match-bad-LPR do poorly, despite being trained with $H_{find}$ which typically result in the opposite effect. This is in fact what we observed: match-good-LPR (blue) substantially outperformed the match-bad-LPR (orange; Figure 7). By constraining LPR, we were able to account for the entire change in performance present in the decoupled find-eval phenomenon, demonstrating that this phenomenon is causally mediated by LPR. Furthermore, the match-good-LPR does even better than the regular pruning runs with good $H_{find}$ from Section 3.1 (green), indicating that the previously "bad" $H_{find}$ were only bad because of their LPR. Interestingly, we observe this same effect when we constrain LPR to be the same for all layers, equivalent to local or uniform layerwise pruning (Figure A15, Figure A16), confirming the importance of LPR to the decoupled find-eval phenomenon.

**Adjusting weight decay and learning rate together** Thus far, we have adjusted each hyperparameter independently, but these parameters can also be adjusted jointly. One particular result of interest is that when the weight decay is set to 0, the learning rate barely affects the LPR (Figure 8; contrast with Figure 6a) – while there is still some visible difference within each block of layers, the larger sections of the network are all pruned equally. Additionally, higher $LR_{find}$ now perform very

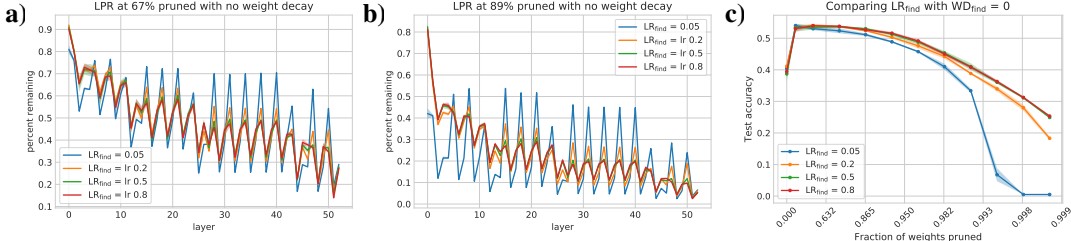

**Figure 8: LPR for different LR$_{\text{find}}$ values at (a) 67% pruned (b) and 89% pruned**. When weight decay is set to zero, LR$_{\text{find}}$ has a significantly smaller effect on LPR. This is in contrast to Figure 6a, where the early layers get pruned much more at higher learning rates. As a result, overall performance is no longer harmed by a large LR$_{\text{find}}$: when comparing a large range of LR$_{\text{find}}$, all evaluated with a reasonable LR$_{\text{eval}}$ = 0.5 and WD$_{\text{eval}}$ = 1e-4, the larger LR$_{\text{find}}$ does better **(c)**.

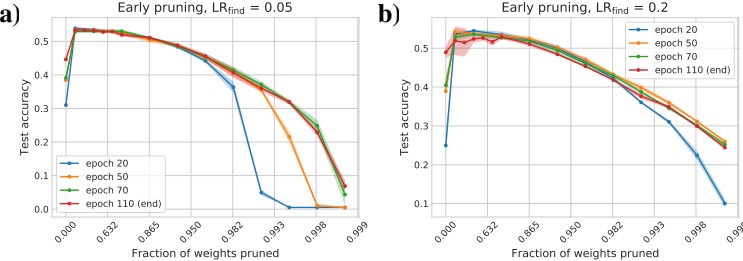

**Figure 9: Early pruning:** Pruning at epoch 20 (blue) is bad, but pruning at epochs 50 (orange) or 70 (green) is actually better than pruning at convergence (110 epochs, red) for LR$_{\text{find}}$ = 0.2. Pruning at 70 for LR$_{\text{find}}$ = 0.05 also performs well. All masks are evaluated with LR$_{\text{eval}}$ = 0.5, except for the points at 0% sparsity, which represent an unpruned model trained for the corresponding number of epochs.

well, even slightly better than moderate or small LR$_{\text{find}}$, similar to the match-good-LPR experiments above. This is in stark contrast to earlier results with a weight decay of 1e-4.

### 3.5 EARLY PRUNING

For a high LR$_{\text{find}}$, LPR changes dramatically over training, while a low LR$_{\text{find}}$ leads to stable LPR throughout training, with the exception of the first layer (Figure A18). Furthermore, the distance between the early mask's LPR and the actual mask's LPR decreases quickly and plateaus, likely due to the learning rate decay (Figure A19). This inspired us to experiment with pruning early, as in Yin et al. (2020). We found that early pruning can be even better than pruning at convergence: pruning at either epoch 50 or 70 for LR$_{\text{find}}$ = 0.2 actually improves performance compared to pruning at the end of training (Figure 9). There are two useful implications here. First, this is evidence of yet another hyperparameter that follows the same phenomenon: the best settings to train a mask for evaluation or to train a standard unpruned model (110 epochs) is not necessarily the same as the best settings to prune the model (50 epochs). Second, for the particular case of epochs spent training, this allows us to save compute. As mentioned in Section 2.1, pruning with decoupled $H_{\text{find}}$ and $H_{\text{eval}}$ requires no extra compute compared to the original LT procedure. In this case, $H_{\text{eval}}$ decreases the compute used to find masks by 55% while also gaining a slight increase in test accuracy.

### 4 ADDITIONAL RELATED WORK

Magnitude pruning, in which weights or filters are removed according to their magnitude, was first introduced in Han et al. (2015), and has had great success, prompting a large field of study, recently summarized by Blalock et al. (2020). Structured pruning approaches have had great success since such techniques can accelerate CNN during test-time without requiring specialized hardware and software support. Li et al. (2016) have demonstrated the potential of channel pruning for improving the efficiency of CNNs at test time. Later, many methods were proposed to further improve the performance of structural pruning (Liu et al., 2017; He et al., 2019b; Molchanov et al., 2019; Fan et al., 2020). Related to our work, Liu et al. (2019) have demonstrated the importance of LPR over pre-trained weights for a wide range of structural pruning methods. Complementing their results,

our findings further suggest that while both weights and LPR are important for pruning techniques, the value of weights is conditioned on good LPR, which is implicitly determined by hyperparameters when it comes to global magnitude pruning. They also pointed out the failure of large LRs for lottery tickets, resulting in lottery tickets being trained with smaller LRs.

The lottery ticket hypothesis argues that over-parameterization is necessary to find a good initialization rather than for optimization itself (Frankle & Carbin, 2018; Frankle et al., 2019a). Since their introduction, a number of works have attempted to understand various aspects of this phenomenon, including their transferability (Morcos et al., 2019), the factors which make particular winning tickets good (Zhou et al., 2019), and linear mode connectivity (Frankle et al., 2019b). The lottery ticket hypothesis has also been challenged by (Liu et al., 2019), in part because of the requirement for small learning rates, as we address in this work.

Tanaka et al. (2020) observed that layer collapse, which they defined as entire layers being completely pruned, is a key issue that causes some pruned models to fail. This is along the same lines of the bad LPR that we observed, with a few key differences: we saw poor performance even before layer collapse, and skip connections in ResNets should make layer collapse less of an issue. Further, they claim that iterative pruning avoids layer collapse, but we see failure cases in iterative pruning.

In Li et al. (2020), the authors also decouple the learning rate between pre-training and fine-tuning, somewhat analogous to how we decouple the learning rate between finding and evaluating a mask; they are similar concepts applied to different tasks. As we analyze the impact of $LR_{find}$, they analyze the significance of the learning rate for fine-tuning on a different dataset.

There are several proposed methods to prune at initialization, which does not require training the model before finding the mask (Lee et al., 2019; Wang et al., 2020; Tanaka et al., 2020). In contrast with our experiments, there is no $H_{find}$ (beyond the initialization distribution), only $H_{eval}$. While pruning at initialization is more efficient resource-wise, Frankle et al. (2020) recently demonstrated in extensive experiments that performance of these approaches is not on par with pruning after training. Hence, it is still critical to study methods that do not prune at initialization.

## 5 DISCUSSION

In this work, we proposed a simple idea: use different hyperparameters for mask discovery and evaluation for pruning. We uncovered the counterintuitive decoupled find-eval phenomenon (Section 3.2), in which using hyperparameters that lead to *worse* model performance yield *better* masks. This effect is consistent across both lottery tickets and structured pruning (Section 3.3). We also demonstrated that values of $H_{find}$ which correspond to effectively small learning rates also lead to materially different LPRs in which early layers are pruned less than later layers (Figure 6a), and that these LPR causally mediate the decoupled find-eval phenomenon (Section 3.4). While LPR is not quite all you need, it does determine a very significant part of the final pruned performance and provides insights into the masks found by pruning.

**Practical implications** Our work also has several key practical implications to improve compressed model performance. Most notably, we found that tuning hyperparameters separately for the two phases of training can result in substantial performance gains. While this does introduce additional hyperparameter dimensions to search over, our experiments suggests several general guidelines for tuning these parameters: $H_{find}$ requires smaller LR and WD or larger BS; $H_{eval}$ is similar to $H_{unpruned}$ at low sparsity levels; and $H_{eval}$ at high sparsity levels should gradually decrease LR and WD or increase BS. Alternatively, we found that simply turning off weight decay during $H_{find}$ produced good LPRs and, consequentially, good performance. Alternatively, don't use weight decay for finding masks and then the LR no longer matters. Finally, we observed that masks generated earlier in training often yielded better overall performance, suggesting that by pre-training for fewer epochs, both compute efficiency *performance* can be improved.

**Limitations and future work** While we showed that the decoupled find-eval phenomenon is causally mediated by LPR, we did not find a clear answer to the question of why certain $H_{find}$ leads to better LPR. In addition, a method to derive a "good" LRP would be very useful. We hope to investigate these question further in future work. Furthermore, it would be interesting to study additional hyperparameters and how they interact, and experiment with more datasets, architectures, and tasks beyond vision classification.

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

# A APPENDIX

## A.1 ADDITIONAL TRAINING DETAILS

---

**Algorithm A1:** Unstructured iterative magnitude pruning

---

**Input:** Model parameters $\boldsymbol{\theta}$, pruning percentage per iteration $p$, number of total pruning iterations $N$, mask-finding hyperparameters $H_{\text{find}}$, mask-evaluating hyperparameters $H_{\text{eval}}$

**Output:** Performance of trained pruned models

1 Initialize mask $M_0$ as all ones
2 Randomly initialize model weights $\boldsymbol{\theta_0}$
3 **for** $i = 1...N$ **do**
4      Train model $\boldsymbol{\theta} \odot M_{i-1}$ to convergence using hyperparameters $H_{\text{find}}$ to obtain $\boldsymbol{\theta_i}$
5      Prune $p\%$ of parameters $\boldsymbol{\theta_i}$, the ones with the smallest magnitudes globally and have not been masked out by $M_{i-1}$, to create a binary mask $M_i$
6      Rewind weights back to initial $\boldsymbol{\theta_0}$
7 **end**
8 Evaluate final mask or other masks of interest ($M_i$) by training $\boldsymbol{\theta} \odot M_i$ with hyperparameters $H_{\text{eval}}$ and record its performance

---

---

**Algorithm A2:** Structured one-shot magnitude pruning

---

**Input:** Model parameters $\boldsymbol{\theta}$, pruning percentage per iteration $p$, mask-finding hyperparameters $H_{\text{find}}$, mask-evaluating hyperparameters $H_{\text{eval}}$

**Output:** Performance of trained pruned model

1 Randomly initialize model weights $\boldsymbol{\theta}$
2 Train model $\boldsymbol{\theta}$ using hyperparameters $H_{\text{find}}$ to obtain $\boldsymbol{\theta}_{\text{final}}$
3 Prune $p\%$ of parameters $\boldsymbol{\theta}_{\text{final}}$ to create a binary mask $M$ by pruning channels with smallest magnitude in the scale for the batch normalization layers, *i.e.*, $\gamma$
4 Evaluate the mask by training $\boldsymbol{\theta} \odot M$ with hyperparameters $H_{\text{eval}}$ and record its performance

---

All unstructured pruning experiments used data augmentation (RandomResizedCrop and RandomHorizontalFlip from torchvision), batch size of 512, and weight decay of 0.0001. We pruned convolutional layers 20% at each pruning iteration, for a total of 30 iterations. Weights were initialized with Kaiming normal distribution.

ResNet-50 on Tiny ImageNet: we trained 110 epochs, decaying LR by gamma = 0.2 at epochs 70, 85, and 100. Experiments with warmup, late rewinding, and learning rate rewinding (Section 3.3) use the same hyperparameters, except for modifications explicitly mentioned in the main text.

ResNet-50 on MiniPlaces: we trained 120 epochs, decaying LR by gamma = 0.2 at epochs 80, 110, and 115.

ResNet-18 on Tiny ImageNet: we trained 110 epochs, decaying LR by gamma = 0.1 at epochs 80 and 100.

Structured pruning: we use the magnitude of $\gamma$ of batch normalization layers as the ranking metric to decide which filters to prune. For channels that are connected by residual connections, we sum their measures as they have to be pruned jointly. We train the pruned network using 200 epochs for both pre-training and fine-tuning. We use a batch size of 256, weight decay of $5e-5$, cosine learning rate decay, linear learning rate warmup from 0 to the set learning rate within the first 5 epochs, SGD with 0.9 Nesterov momentum, and 0.1 label smoothing. All layers were initialized with Kaiming normal distribution. The algorithm is specified in A2. Note that we do one-shot pruning and do not rewind weights back to their initial values, contrary to the method for unstructured pruning.

Shaded error bars in all figures represent standard deviations over 5 runs with different random seeds.

## A.2 ADDITIONAL FIGURES REFERENCED FROM SECTION 3

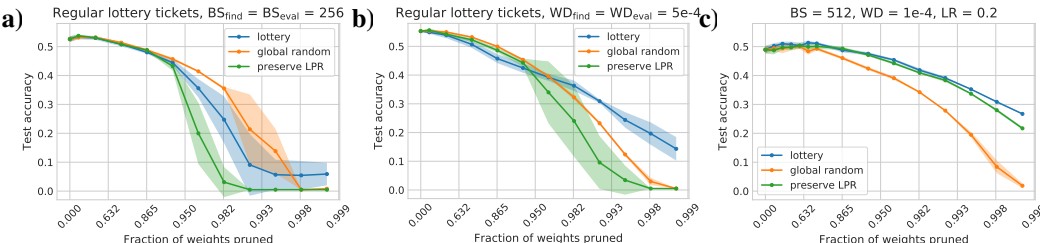

**Figure A1:** Test accuracy at various sparsity levels, with the regular lottery ticket procedure, i.e. $H_{find} = H_{eval}$. Smaller batch size **(a)** and larger weight decay **(b)** result in poorly performing lottery tickets, despite being better for the regular unpruned model, represented by the points at 0% sparsity. **(c)** is the same as Figure 1b, copied over for comparison, which uses the default hyperparameters (but not the best ones for training a regular model).

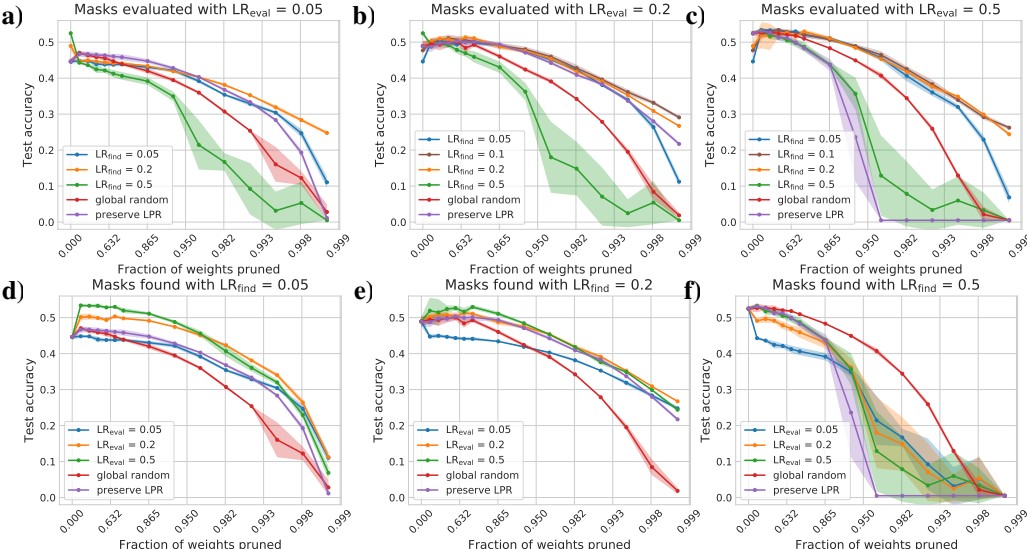

**Figure A2:** All combinations of $LR_{find}$ and $LR_{eval}$. Top **(a-c)**: given a constant $LR_{eval}$, we compare different values of $LR_{find}$. 0.1 does the best, followed closely by 0.2 then 0.05, and everything is significantly better than 0.5. Bottom **(d-f)**: given a constant $LR_{find}$, we compare different values of $LR_{eval}$. 0.5 does the best at lower sparsity levels and 0.2 at higher sparsity levels. Random baselines have the same $LR_{find}$ and $LR_{eval}$ as the value in the title.

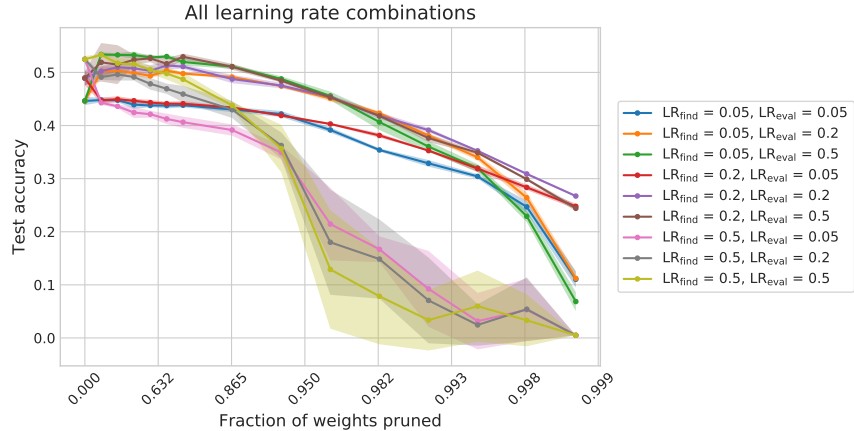

**Figure A3:** All combinations of learning rate plotted together for comparison

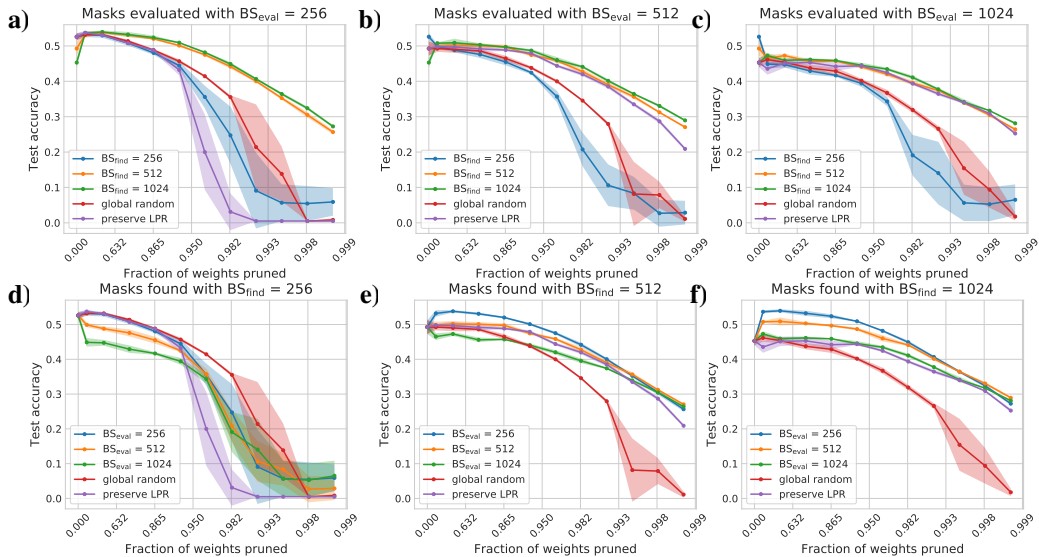

**Figure A4:** All combinations of $BS_{find}$ and $BS_{eval}$. Top **(a-c)**: given a constant $BS_{eval}$, we compare different values of $BS_{find}$, showing that larger is better. Bottom **(d-f)**: given a constant $BS_{find}$, we compare different values of $BS_{eval}$. 256 does the best at low to medium sparsity levels and 512 at higher sparsity levels. Random baselines have the same $BS_{find}$ and $BS_{eval}$ as the value in the title.

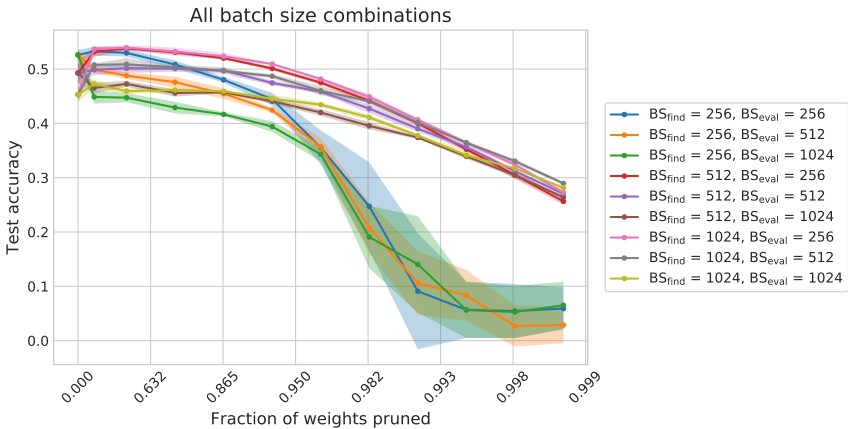

**Figure A5:** All combinations of batch size plotted together for comparison

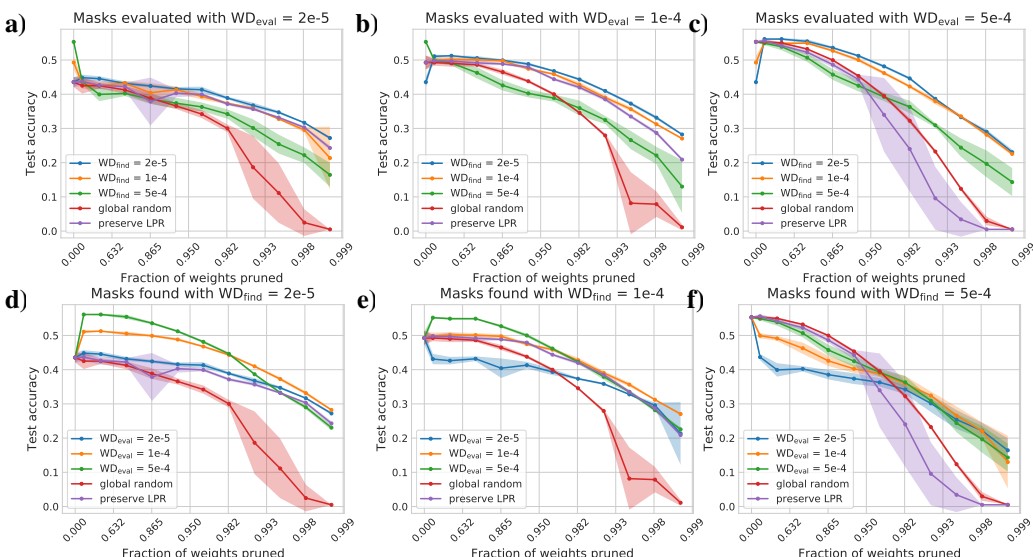

**Figure A6:** All combinations of $WD_{find}$ and $WD_{eval}$. Top **(a-c)**: given a constant $WD_{eval}$, we compare different values of $WD_{find}$, showing that smaller is better. Bottom **(d-f)**: given a constant $WD_{find}$, we compare different values of $WD_{eval}$. 5e-4 does the best at lower sparsity levels and 1e-4 at higher sparsity levels. Random baselines have the same $WD_{find}$ and $WD_{eval}$ as the value in the title.

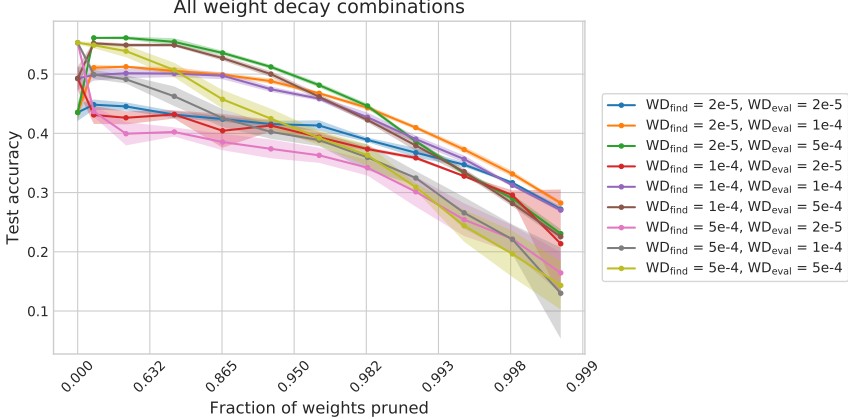

**Figure A7:** All combinations of weight decay plotted together for comparison

### A.2.1 OTHER MODELS AND DATASETS

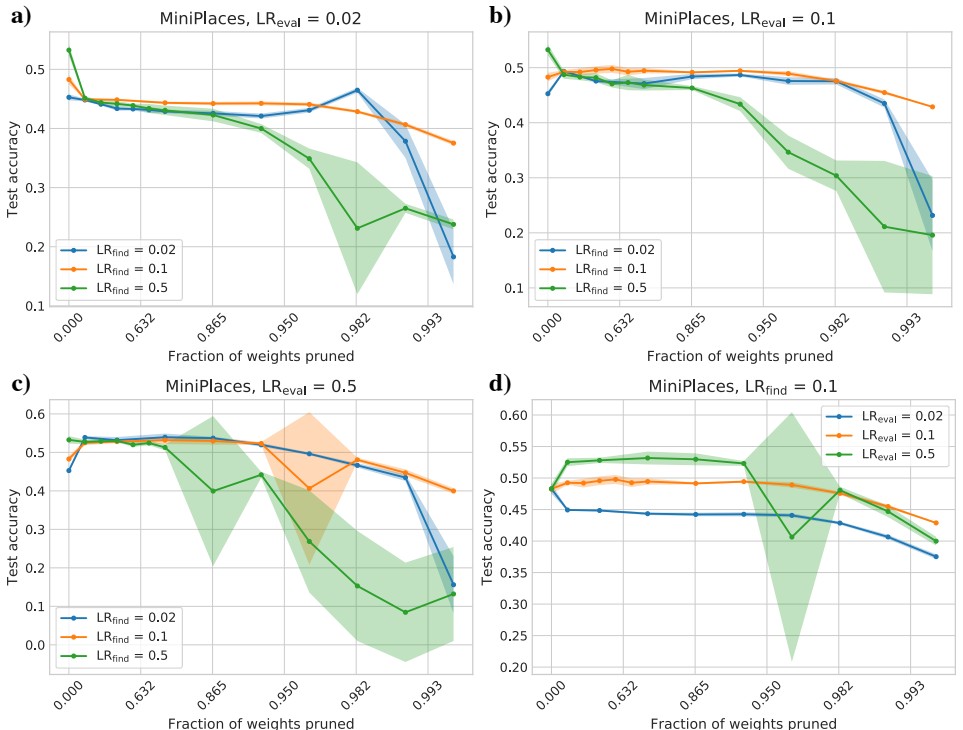

**Figure A8:** ResNet-50 on MiniPlaces with $LR_{find}$ and $LR_{eval}$ from $\{0.02, 0.1, 0.5\}$. In (a), (b), and (c), we use a constant $LR_{eval}$ of 0.02, 0.1, and 0.5 respectively, with each plot showing all three values of $LR_{find}$. They show that $LR_{eval} = 0.1$ is the best. Thus, in (d) we compare the different values of $LR_{eval}$, all with $LR_{find} = 0.1$, to demonstrate that the best $LR_{eval}$ is 0.5, until very high sparsity levels where 0.1 becomes better. In line with other results, the best LR for standard training (0.5) is not the best $LR_{find}$ for pruning.

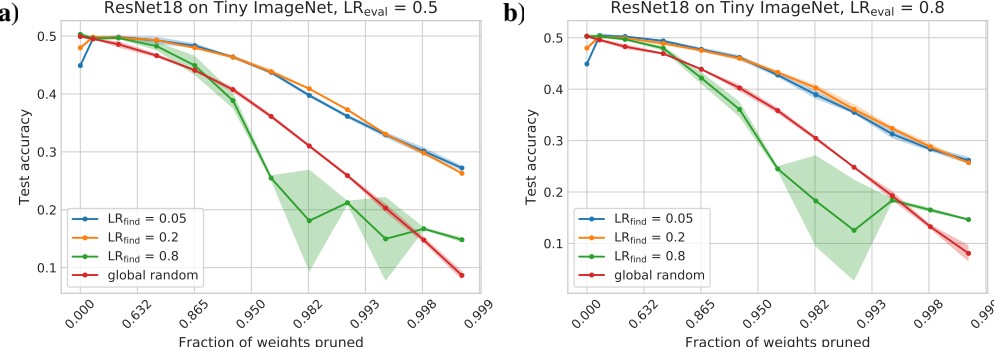

**Figure A9:** ResNet-18 on Tiny ImageNet with $LR_{find} \in \{0.05, 0.2, 0.5\}$, compared using a constant $LR_{eval}$ of 0.5 in (a) and 0.8 in (b). For both values of $LR_{eval}$, $LR_{find}$ of 0.05 and 0.2 are very similar and both perform significantly better than $LR_{find} = 0.8$, despite 0.8 performing the best on the unpruned model. $LR_{find}$ of 0.8 also does worse than the random baseline (red).

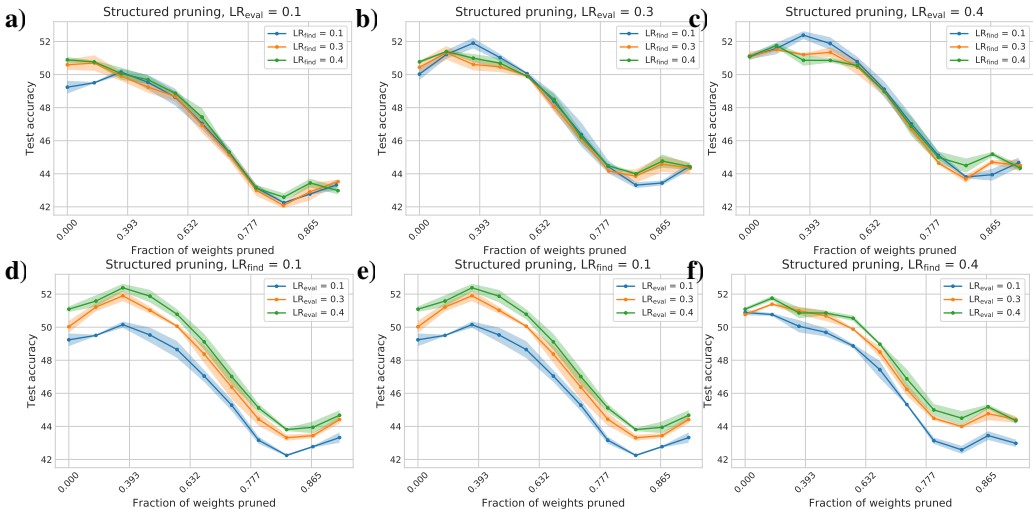

**Figure A10:** Structured pruning: All combinations of $LR_{find}$ and $LR_{eval}$. Top **(a-c)**: given a constant $LR_{eval}$, compare different values of $LR_{find}$. Bottom **(d-f)**: given a constant $LR_{find}$, compare different values of $LR_{eval}$.

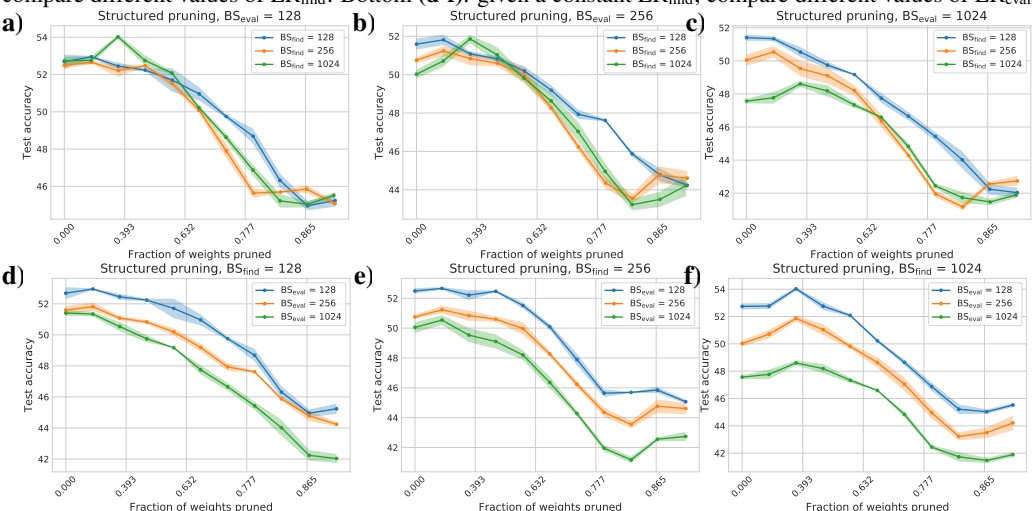

**Figure A11:** Structured pruning: All combinations of $BS_{find}$ and $BS_{eval}$. Top **(a-c)**: given a constant $BS_{eval}$, compare different values of $BS_{find}$. Bottom **(d-f)**: given a constant $BS_{find}$, compare different values of $BS_{eval}$.

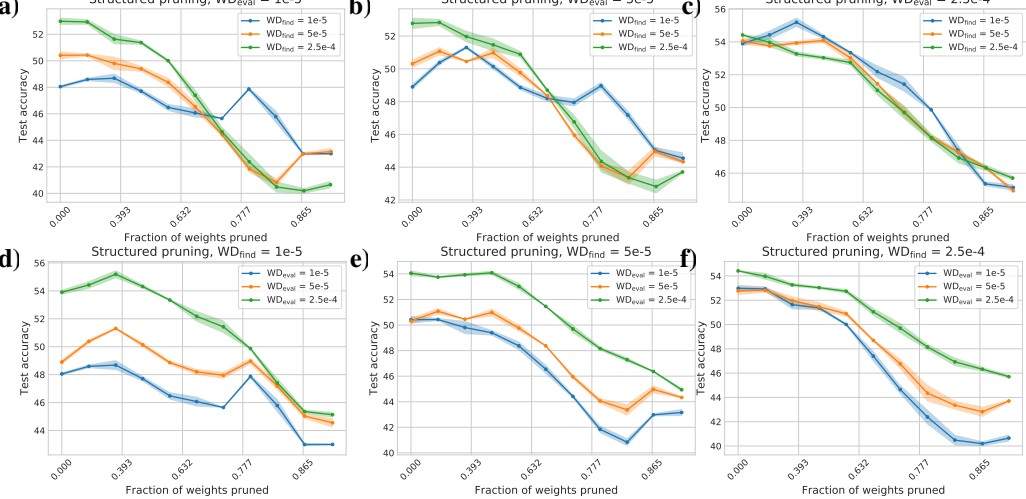

**Figure A12:** Structured pruning: All combinations of $WD_{find}$ and $WD_{eval}$. Top **(a-c)**: given a constant $WD_{eval}$, compare different values of $WD_{find}$. Bottom **(d-f)**: given a constant $WD_{find}$, compare different values of $WD_{eval}$.

### A.2.2 OTHER VARIANTS

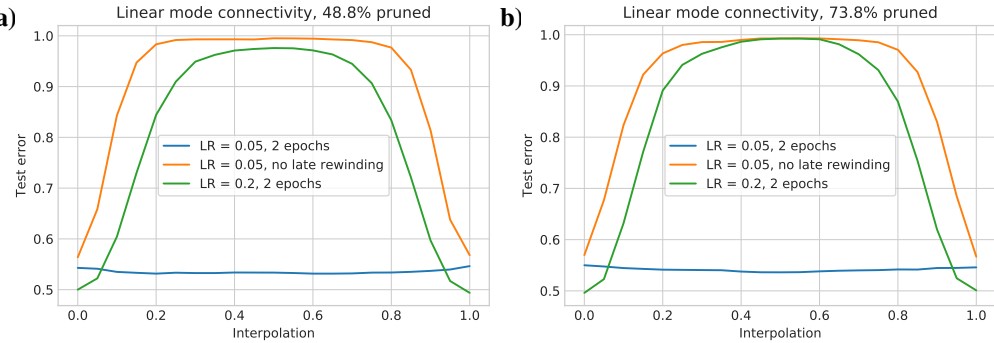

**Figure A13: Instability analysis from Frankle et al. (2019b) of various networks.** We measure the mode connectivity between two different networks trained with the same initialization but different SGD noise. Plotted is the test error as we linearly interpolate between the two networks, for (a) 48.8% pruned and (b) 73.8% pruned. Using LR = 0.05 (both $LR_{find}$ and $LR_{eval}$, as per a regular LT run) starting from 2 epochs, the network is stable, thus late rewinding of 2 epochs is reasonable. In contrast, LR = 0.05 without late rewinding (orange) and a larger LR = 0.2 from 2 epochs (green) both are not stable.

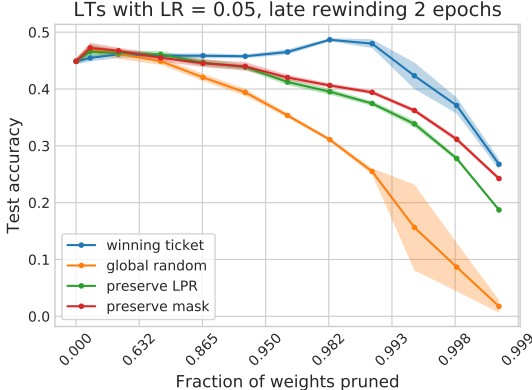

**Figure A14:** Regular lottery tickets, with $LR_{find} = LR_{eval} = 0.05$ and late rewinding to 2 epochs, compared to random baselines. The winning ticket does significantly better than all random baselines, including "preserve mask", which is a random baseline that uses the same mask as the winning ticket but with reinitialized weights. This indicates that the network is stable at 2 epochs (Frankle et al., 2019b).

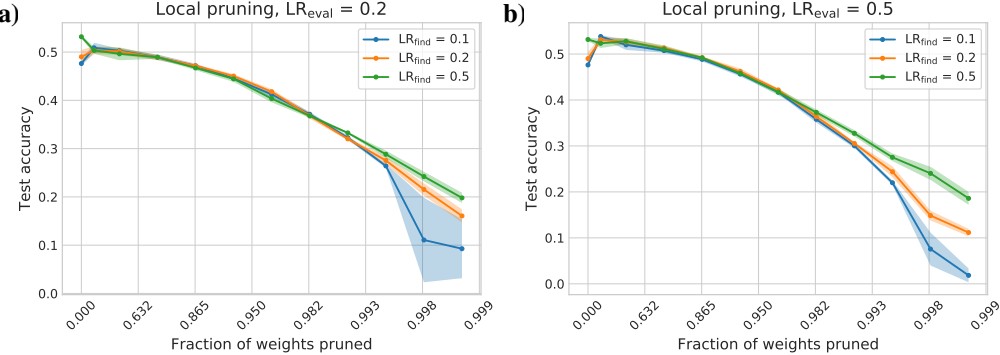

**Figure A15:** Local pruning: pruning all layers the same amount. We compare $LR_{find} \in \{0.1, 0.2, 0.5\}$ using $LR_{eval}$ of 0.2 (a) and 0.5 (b). For both values of $LR_{eval}$, we see that the larger the $LR_{find}$, the better the pruned models perform.

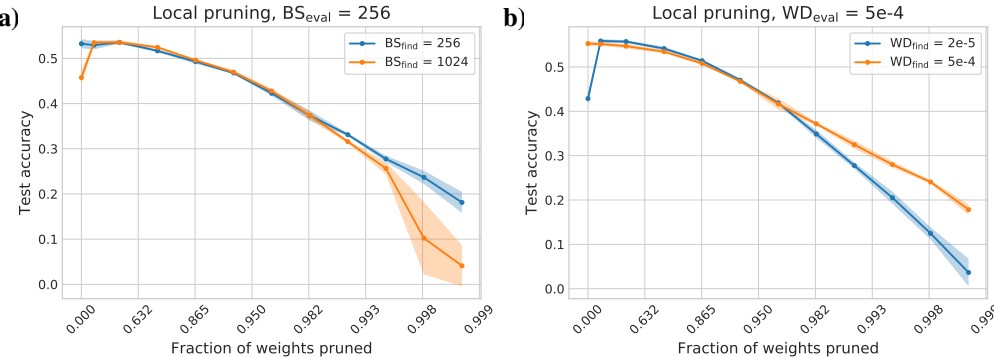

**Figure A16:** Local pruning: pruning all layers the same amount. Similar experiment setup as Figure A15, except comparing different values of $BS_{find}$ and $WD_{find}$. While large $BS_{find}$ and small $WD_{find}$ performed well in global pruning, we see the opposite for local pruning.

### A.2.3 ADDITIONAL LPR FIGURES

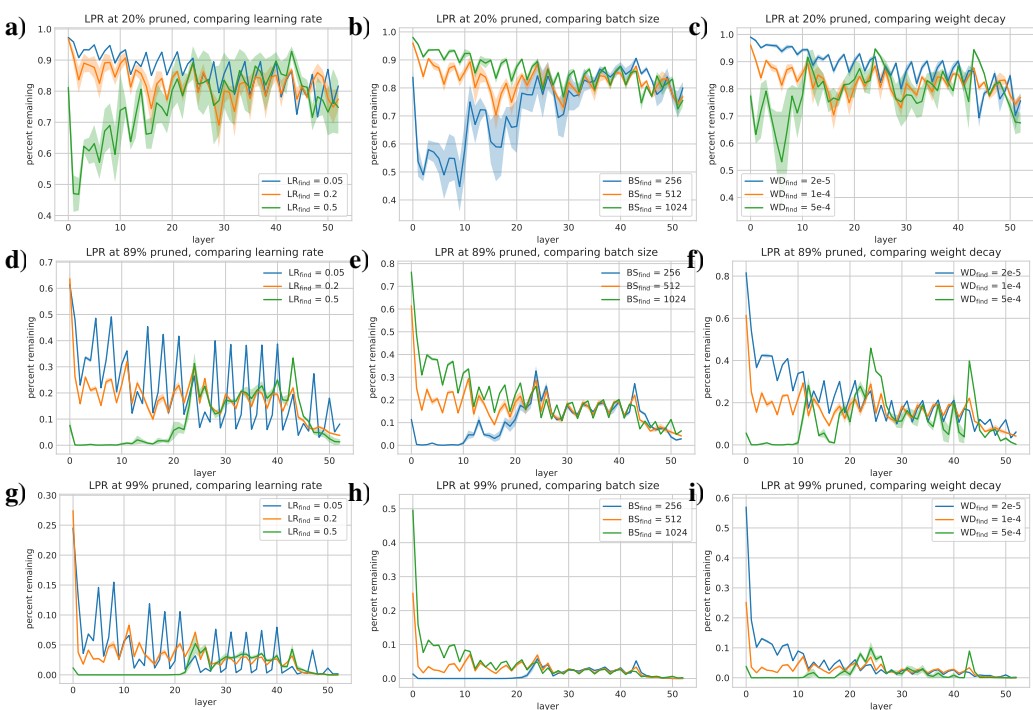

**Figure A17:** Comparison of LPR of masks found by the three different values of $LR_{find}$ (**a**), $BS_{find}$ (**b**), and $WD_{find}$ (**c**) at 20, 89 and 99% sparsity (1, 10, and 20 pruning iterations respectively). Similar to Figure 6 at 67% sparsity (5 pruning iterations), small LR, large BS, and small WD values tend to prune more in the earlier layers.

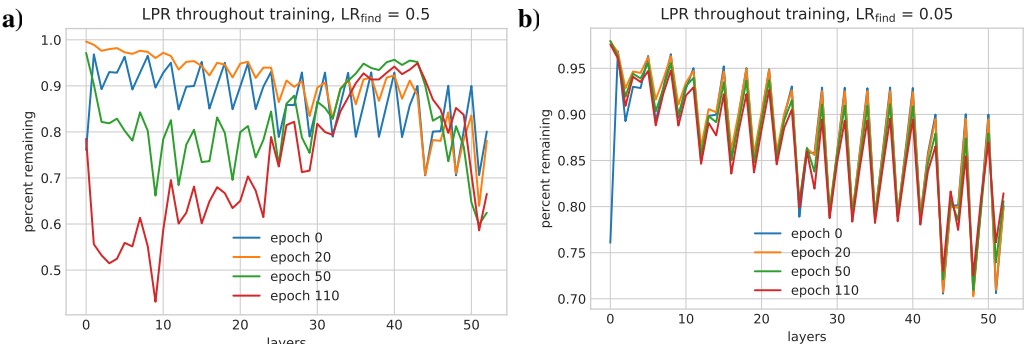

**Figure A18:** Layerwise pruning ratios if you were to prune at different epochs, for $LR_{find} = 0.5$ (a) and 0.05 (b), at the first pruning iteration (starting from an unpruned model). In (a), the LPR change significantly throughout training, whereas for (b), LPR changes very slightly except for the first layer.

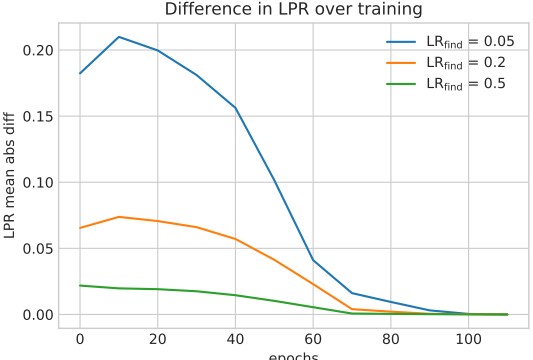

**Figure A19:** Distance between LPR if pruned at epoch $n$ vs. actual LPR when pruned at the end, for different $LRs_{find}$. Distance is calculated as the mean absolute difference in pruning ratio for each layer. This is done at the first pruning iteration (starting from an unpruned model). Smaller $LR_{find}$ values start with much smaller distances; all $LR_{find}$ values start to plateau around epoch 70, which is when the learning rate decays.

