# OpenReview forum: "Uncovering the impact of hyperparameters for global magnitude pruning"
_ICLR.cc/2021/Conference — Reject_

### Official Review · AnonReviewer4 · 2020-10-21
**I don't think the assumptions the authors made about current practices, which are the motivation for the propose method, are correct.**

**Rating:** 4
**Confidence:** 5

**Review:**

### Summary:
The authors propose to use a different learning rate (LR) when fine-tuning a pruned model (LR_eval) than the learning rate used for the original training (LR_find). They empirically demonstrate in a Lottery Ticket (LT) framework, using a ResNet50 on Tiny ImageNet, that the LR that produced the best model before pruning does not correspond to the LR that produces the best results after fine-tuning. They then further show this phenomenon on other LT setups (with late rewinding, LR warm-up, LR rewinding), architecture (ResNet18), and dataset (MiniPlaces), and for structured pruning as well. Finally, they empirically show how LR_find impacts the pruning ratio obtained at each layer, and that high LR_find seems to prune too aggressively the first layers.

### Strengths:
+ The paper tries to provide better understanding of pruning methods, rather than introducing a new heuristic, which is an extremely important research direction in network pruning.
+ The results are presented on different architecture, datasets, LT variants, and on both structured and unstructured pruning setups.

### Concerns:
+  I have issues the motivation behind the proposed method. In particular, these few sentences (listed below) sound quite incorrect to me, at least in the LT framework, which is the framework used for in this empirical evaluation:
   - _"However, prior work has implicitly assumed that the best training configuration for model performance was also the best configuration for mask discovery."_
   - _"Common practice rests on the assumption that models with the best performance will generate the best masks, such that the optimal hyper-parameters for mask generation and mask evaluation should be identical."_
  - _"If we had to use the same LR for everything, we would have to make a trade-off between a better unpruned performance and a better pruned model performance."_
  - _"If we had followed common practice in the past, we would have swept over some LR values over standard training, found that 0.5 performed the best, trained LT with 0.5 as the LR\_find, and gotten poor performance."_

  In a pruning scenario where we have to both train the network and then prune it, like in the LT framework, the hyper-parameters are optimized to produce the best performing models after the iterations of pruning and fine-tuning, not the best performing network before the first pruning!
  Ideally, one would have to tune all the hyper-parameters, which means the hyper-parameters at each of the LT iterations, not only the ones of the last iteration as proposed, which is prohibitively expensive. However Renda et al. (2020) showed that using the same ones at each iteration is a good heuristic, and this is precisely what the experiments of this paper seems to show (the best overall results are: in Figure (2) LR_eval = LR_find = 0.2, in Figure (3) LR_find = LR_eval = 0.2, in Figure (A1) LR_find = LR_eval = 0.1, etc...). I do agree however that one should not first tune the hyper-parameters to obtain the best network, and then reuse these same hyper-parameters for the pruning stages, and your experiments indeed show that, but I don't think this is what practitioners do in practice.

+ While the analysis of how the pruning ratio per layer changes depending on LR_find is quite interesting, it lacks a bit of insights. What is causing this? Are other pruning criteria also affected? One can imagine the neural networks trained with high learning rate have gradient propagation issues (Magnitude Pruning removes the smallest weights, so the ones that received the smallest gradient). On that note, how are the networks initialized (this information is missing from the document)?

### Reasons for score:
As it stands, I don't think this paper brings much new insights, and I don't think the assumptions the authors made about current practices, which are the motivation for the propose method, are correct.

### Questions:
- At each iteration of LT, what is the model selected to compute the mask on? Is it the model obtained at the last iteration, or the model that was best performing on the validation set? I imagine this could have a big impact on the results, as a high learning rate might produce models drastically different.

### Other comments:
There are a few inaccuracies in the paper:
 - (2.1) what are one shot and local pruning (it should be defined in the paper)?
 - What are error bars representing in the figure? Standard deviation? Standard error? How many seeds were used? This should be precised in the document.
 - (2.2) What does "decaying the learning rate with a gamma of 0.2" means? Again, the type of decay used should be precised.

Finally, Footnote 1 is quite unnecessary.

---

> ### Author Response · Authors · 2020-11-20
> **Thank you for the review; here is our response**
>
> **Response to concerns:**
>
> Assumption about current practices are incorrect:
> - We agree that the narrative was not stated as clearly as it could have been. In particular, we neglected to clearly state that hyperparameters are often optimized separately for the lottery ticket procedure such that H_unpruned /= H_LT. We have added direct statements to clarify that this has been done in prior work in Sections 2.1 and 3.1.
> - However, we note that this is not the central motivation for our work. Critically, our work does not focus on the fact that H_unpruned /= H_LT, but rather that for H_LT, the same hyperparameters were used for both mask discovery (H_find) and mask evaluation (H_eval). Our work demonstrates that decoupling improves performance substantially (by 3-4% at most non-extreme sparsities) and demonstrates that this effect is mediated causally by layerwise pruning ratios.
>
> “However Renda et al. (2020) showed that using the same ones at each iteration is a good heuristic, and this is precisely what the experiments of this paper seems to show (the best overall results are: in Figure (2) LR_eval = LR_find = 0.2, in Figure (3) LR_find = LR_eval = 0.2, in Figure (A1) LR_find = LR_eval = 0.1, etc...).”
> - We acknowledge that within Figure 2b (of the original submission; Figure A2b in revised submission), LR_find = LR_eval = 0.2 is best. However, we emphasize that the appropriate comparison in Figure A2 is not *within* each plot, but across all three plots a-c.
> - Nevertheless, we readily acknowledge that this presentation is confusing since it does not allow direct comparison. We have therefore added Figure 3 to the revised paper, which compares the best decoupled hyperparameters to the best coupled hyperparameters for learning rate, batch size, and weight decay. In all cases, the best decoupled setting is better than the best coupled setting, with gaps of 3-4% below 98% sparsity.  To allow detailed comparisons across individual combinations, we have also added Figures A3, A5, and A7, which show all combinations for learning rate, batch size, and weight decay, respectively.
> - We also note that since the initial submission, in addition to including batch size and weight decay, we also evaluated additional learning rates, finding that LR_find = 0.1 is substantially better than both LR_find = 0.05 and LR_find = 0.2.
>
> “While the analysis of how the pruning ratio per layer changes depending on LR_find is quite interesting, it lacks a bit of insights. What is causing this? Are other pruning criteria also affected? One can imagine the neural networks trained with high learning rate have gradient propagation issues (Magnitude Pruning removes the smallest weights, so the ones that received the smallest gradient).
> - We agree that the question of how learning rate, batch size, and weight decay impact LPR is an extremely interesting one. However, addressing this question is non-trivial and we would argue that it is beyond the scope of the present work, especially given how extensive it already is (for example, our revised submission already has 19 appendix figures).
>
> “How are the networks initialized (this information is missing from the document)?”
> - Networks are initialized with Kaiming normal distribution. Thank you for pointing out that this is missing; we have added it to the training details section.
>
> **Response to other questions and comments:**
>
> “What is the model selected to compute the mask on? Is it the model obtained at the last iteration, or the model that was best performing on the validation set? “
> - For the main experiments, we used the model obtained at the last iteration, as done in [“One ticket to win them all”, Morcos et al 2019, “Comparing rewinding…”, Renda et al 2020]. This is not very different from the model with the best validation accuracy, since the validation performance does not drop noticeably. In the early pruning section, we prune at points earlier in training, but that is based on a set number of epochs rather than chosen based on validation accuracy.
>
> “What are one shot and local pruning”:
> - We have updated our descriptions to make this clearer the first time they are mentioned. One shot pruning is defined in Algorithm A2 (for structured pruning), where you only train once, prune, and train the pruned model. This is different from iterative pruning, where you will prune and retrain multiple times. Local (or uniform layerwise) pruning means that you prune n% of weights each layer, rather than the n% lowest magnitude weights over the whole network. We apologize for the omission.
>
> What are error bars representing in the figure?
> - All error bars represent standard deviation over 5 independent replicates. This was originally stated only in the appendix, but we have added it to the main paper for clarity.
>
> Decaying learning rate:
> - We used a step decay schedule for learning rate decay, meaning that the learning rate was multiplied by a factor of gamma (0.2) at the specified iterations.

---

> > ### Comment · AnonReviewer4 · 2020-11-20
> > **Response to the response**
> >
> > Dear authors,
> >
> > Thank you for taking the time for addressing my concerns and comments about the paper. Figure 3 indeed better highlights the advantages of performing proper hyper-parameter optimization. Thank you as well for the information regarding initialization, model selection, error bars and decay schedule.
> >
> > However, these comment do not change my rating. I still find that the take-home message of the paper, which is to decouple the hyper-parameters for training from the ones to find the mask, is not substantial enough. As I mentioned in my previous comment, we know that we should tune as many hyper-parameters as possible. For the lottery ticket setup, where each iteration can be seen as both an '_eval' and a '_find', one would have to tune the learning rate, batch size, weight decay (etc) for each iteration of the lottery. This comes at a high computational cost (as you can probably see when scaling up to ImageNet) that practitioners will typically not afford to pay, and will thus likely fall back to the recommendations from Renda et al 2020.

---

### Official Review · AnonReviewer1 · 2020-10-28
**Paper finding that hyperparameters that lead to more accurate networks do not necessarily lead to better pruned networks**

**Rating:** 7
**Confidence:** 5

**Review:**

# Summary

This paper evaluates explicitly decoupling the hyperparameters (specifically, learning rate) used to find pruning masks from those used to train pruned networks, finding that for global magnitude pruning lower learning rates (relative to the learning rate that results in the most accurate full-size network) result in masks that can train to higher accuracies; on the other hand, higher learning rates tend to train the pruned networks better. These results are shown to generalize across pruning and re-training techniques (including the standard lottery ticket regime, lottery tickets with warmup, weight rewinding, learning rate rewinding, and structured pruning). The paper then shows that these differences in performance are primarily due to differences in layerwise pruning rates that come from training with different learning rates.

# Strengths

- The paper proposes an interesting experiment, decoupling mask-finding hyperparameters from mask-training hyperparameters, with the potential to change how people think about network pruning, challenging the assumption that a higher-accuracy full network results in a higher-accuracy pruned network
- The findings are well evaluated, convincingly showing that lower learning rates than those that result in the highest-accuracy full network result in better masks
- The analysis is also strong, showing that the primary factor in the performance delta is the layerwise pruning rates

# Weaknesses

- Given that many of the findings are centered around performance when resetting the weights to the beginning of training, which is known to not work with large-scale networks, a significant amount more discussion of or replications of experiments of [1] is warranted.
- Presentation of figures could also be significantly improved: for instance, by moving figures closer to text that references them, by collating high-level takeaways into tables, and otherwise making it easy to process the large amount of data in this paper
- The early pruning subsection seems to refer to results that are not presented in the paper; given that the data is presented in the appendix, it'd be better to include this text in the appendix.

# Overall recommendation

7: Accept

# Other comments and suggestions

- The original training hyperparameters could be much more clearly presented in the main body of the paper
- It would be helpful to show the original accuracy of the fully trained networks (at various different learning rates) on the plots

# References used in review:

[1] Jonathan Frankle, Gintare Karolina Dziugaite, Daniel M. Roy, and Michael Carbin. "Linear Mode Connectivity and the Lottery Ticket Hypothesis"


# Update after author response

Thanks to the authors for their response. I appreciate the inclusion of Figure A13,  though more details (specifically, whether the plot for LR=0.2 is both LR_find and LR_eval, as I suspect but can't confirm) and more runs (specifically, the decoupled LR lines in Fig 4a that outperform LR=0.05, probably the LR_find=0.05 and LR_eval=0.2) would be appreciated.

I don't agree with the authors' claims about standard LT (rewinding to iteration 0) being equally as valid of an object of study as rewinding to later iterations on these large-scale networks, because by the original definition of lottery ticket (a sparse randomly initialized subnetwork that matches the accuracy of the full network in the same amount of training time), there do not exist standard lottery tickets on ResNet-50 at nontrivial sparsities using the standard learning rate schedules, and the resultant "lottery ticket" network trains little better than a random subnetwork [1, Figure 10], implying that LR_find may not actually be a relevant hyper-parameter in this large-scale rewind-to-0 context. I still think more discussion of this point is also warranted when discussing results on ResNet-50 when rewinding to iteration 0.

Regardless, I believe that the paper presents and thoroughly validates an interesting hypothesis, and I maintain that the paper should be accepted.

---

> ### Author Response · Authors · 2020-11-20
> **Thank you for the review; here is our response**
>
> “Given that many of the findings are centered around performance when resetting the weights to the beginning of training, which is known to not work with large-scale networks, a significant amount more discussion of or replications of experiments of [1] is warranted.”
> - As suggested by R2, we have run additional analyses to confirm that the late rewinding setting we examined is valid. Specifically, we included new linear mode connectivity analyses in Figure A13, following [1; Frankle et al., ICML 2020]. These results show that late rewinding to epoch 2 is indeed stable, verifying that our late rewinding results do apply to the stable phase of training.
> - More broadly, however, within the lottery ticket framework, there is the original procedure where after pruning, weights are rewound back to their original initializations. Only in followup papers is late rewinding used. Both the original (rewind to iteration 0) and the late rewinding framework (rewind to iteration > 0) are valid methods; while late rewinding works better in some cases, the actual lottery ticket hypothesis (the claim that the original network contains a subnetwork that can do well on its own) requires rewinding to initialization, and ideally, LTs should work without late rewinding, which substantially confuse the scientific understanding of lottery tickets by introducing data-dependence into the rewound weights. Thus, we argue that it is valuable to study both late rewinding and rewinding to init.
>
> Presentation of figures, early pruning subsection, table to summarize high level takeaways:
> - Thank you for the suggestions, we have re-positioned figures as best as we could and moved the early pruning figure to the main text using the additional page allowed at rebuttal. We have also added Figure 3 to help clearly summarize the difference between the best coupled and decoupled runs.
>
> “It would be helpful to show the original accuracy of the fully trained networks (at various different learning rates) on the plots”
> - That is an indeed a helpful comparison, and we show that as the leftmost points at 0% sparsity. The points have the same H_unpruned as the H_find of the rest of the line.

---

### Official Review · AnonReviewer2 · 2020-10-28
**Interesting observations about pruning but open to many conflicting interpretations**

**Rating:** 4
**Confidence:** 5

**Review:**

## Summary

The paper studies the effect of different learning rates on finding performant pruning masks that can be applied to train a sparsified neural network in a lottery-ticket-style framework (i.e. train the original random initialization with the resulting pruning mask applied). The study hereby focuses on a particular setting: find a pruning mask using iterative global magnitude pruning (IMP) applied to various ResNets trained on Tiny ImageNet. The paper finds that a small learning rate is beneficial in finding a performant pruning mask, although the resulting network's performance may be worse, while a large learning rate is better suited to then optimize the sparsified network in the subsequent lottery-ticket-style training procedure. The authors further hypothesize that the per-layer prune ratios (LPRs) found by small learning-rate IMP masks are the main driving factor in increasing the performance of the sparse network in the subsequent training process. This is corroborated with a range of experiments where the LPRs from small learning rate IMP pruning masks is used to find masks (with pre-specified LPRs) using larger learning rates.


## Score

I enjoyed reading the paper and found the resulting conclusions quite interesting. As pointed out by related work before [1, 2, 3] the learning rate can have a huge impact on the performance of lottery tickets and related experimental settings (such as lottery-ticket-style masks with random re-initialization of the weights). This paper provides additional experimental evidence for this phenomenon by separating the effects of training hyperparameter on the mask-finding procedure and the sparse-training procedure.

However, I am not entirely convinced about the framing of the paper itself. I believe the experiments are valid and have merit on their own but in my opinion the authors mix two related but distinct concepts, namely:
1.  Lottery Tickets (LTs): Pruning masks that occur (in the general setting as pointed out by [2, 3]) _early in training_ and _not at initialization_. Specifically, the LT hypothesis in the more general setting states that if the experiment is repeated _exactly the same_ starting from that early iteration in training but with the applied pruning mask, it does not harm the performance of the resulting network.
2. Pruning at initialization: Inspired by LT [4, 5, 6] (but also concurred work [7] with LT), various authors have proposed pruning methods that sparsify the network at the time of random initialization and then train the resulting, sparsified network. These methods usually perform better than when the network is pruned uniformly at random.

Consequently, the conclusions that are drawn in this paper are somewhat confusing, it is really hard to discern the generality of the results, and to understand what is actually observed. Are these observations related to LTs or are the authors proposing another method, i.e. small learning rate IMP, to find performant pruning masks at initialization?

I think that must be clarified before this paper can be accepted in order to for the paper to be beneficial to the community as a whole and to ensure that the observations are not just spurious effects of the particular experimental settings.

## Ways to Improve My Score

Mainly, I think the paper must be properly contextualized in a clear setting that either studies LTs in general or studies pruning at initialization. Right now it sits in the middle and thus it hards to draw appropriate conclusions as the reader of the paper. Specifically, I can see two avenues for the paper to improve its framing of the results:
1. Show that the conclusions hold for a more _general_ LT setting. That is, use a version of Algorithm 1 that rewinds to a stable training phase, c.f. [3, 8], where it has been previously shown that valid LTs can be found using standard IMP. As a result that would naturally require the authors to repeat the experiments.
2. Frame the procedure (small learning rate IMP) as another way to perform pruning at initialization, which performs well. That would require the authors to compare to other pruning at initialization methods [4-7] to understand the performance. Also this would raise the question of why one would use the method in the first place since it is computationally much more expensive than the methods of [4-7].

I have additional feedback in the "Weaknesses" section as discussed below. The minor feedback won't necessarily change my score but I believe can help you strengthen the paper upon publication.

## Strengths

* I commend the authors for a very clean and well-written paper. It is easy to follow, well-structured, and provides sufficient context.

* The experimental study seems to be carried out at a high level. Hyperparameters are clearly summarized and the provided level of detail is sufficient to reproduce the experiments.

* Each plot is clearly labeled and contains shaded error region (i.e. experiments were repeated multiple times). Plots are also all well-structured and easy to interpret.

* The study on the effect of separate hyperparameters for sparse training and discovering sparse masks could be helpful in guiding future research on pruning.

* The part about LPR being the main driven factor for the improved performance is really interesting.

## Weaknesses

On top of the concerns I have previously raised, here are some additional points of feedback:

* What is the reason to mostly rely on Tiny ImageNet for drawing the conclusions? Most pruning work considers CIFAR and ImageNet. Since the conclusions of the paper are dependent upon previous papers it would be helpful to consider the inclusion of at least ImageNet.

* Why do the late rewinding experiments use epoch 2 as rewind epoch? I believe this is misleading. As stated in [3, 6] the "stable phase" of training where LTs can be observed in a general setting usually don't occur until later in the training. So these experiments don't really add value since epoch 2 is most likely not in the stable phase and so it might be confusing to the reader rather than helpful.

* This is mostly related to my main points raised in the "Score" section. Since all of these experiments consider rewinding to an unstable phase of training (i.e. back to the random initialization or a very early in training), I am really uncertain whether these conclusions hold for LTs or pruning at initialization. To me personally, the main conclusion is that if I want to use IMP for finding a pruning mask at initialization, I should use a small learning rate. But then again. Why would I use IMP to find a pruning mask at initialization if there is more efficient methods [4-7] for that?

* The work of [9] is drawing somewhat analogous conclusions about the importance of the distribution of LPRs for pruning at initialization. In particular, the authors conclude that mimicking the LPRs found during pruning at initialization with various methods is essentially sufficient to reproduce the accuracy of the resulting sparsely trained network. I couldn't find a version of [9] in a peer-reviewed venue, so it might count as concurrent submission but nonetheless I think it is crucial to compare the results.

## Other Minor Feedback

* Please clarify in the introduction that all observations are entirely limited to magnitude pruning. I don't think that any conclusions can be drawn for pruning in general and some of the text may hint at a more general phenomenon.

* You could add the appendix directly to the main document. It will be easier to read and jump back and forth between the main body and the appendix.

## References

1. [Rethinking the Value of Network Pruning](https://openreview.net/forum?id=rJlnB3C5Ym)
2. [The State of Sparsity in Deep Neural Networks](https://arxiv.org/abs/1902.09574)
3. [Linear Mode Connectivity and the Lottery Ticket Hypothesis](https://arxiv.org/abs/1912.05671)
4. [Picking Winning Tickets Before Training by Preserving Gradient Flow](https://openreview.net/forum?id=SkgsACVKPH)
5. [Pruning neural networks without any data by iteratively conserving synaptic flow](https://arxiv.org/abs/2006.05467)
6. [Progressive Skeletonization: Trimming more fat from a network at initialization](https://arxiv.org/abs/2006.09081)
7. [Snip: Single-shot network pruning based on connection sensitivity](https://openreview.net/forum?id=B1VZqjAcYX)
8. [Comparing Rewinding and Fine-tuning in Neural Network Pruning](https://openreview.net/forum?id=S1gSj0NKvB)
9. [Pruning Neural Networks at Initialization: Why are We Missing the Mark?](https://arxiv.org/abs/2009.08576)

---

> ### Author Response · Authors · 2020-11-20
> **Thank you for the review; here is our response**
>
> **Response to concerns:**
>
> Clarifications on the framing of the paper, and mixing the two concepts of LTs with late rewinding vs. pruning at initialization:
> - We emphasize that we do not investigate pruning at initialization at all. While a very interesting topic, pruning at initialization is purely aimed at saving compute and does not require any training before pruning. As such, there is no way to decouple H_find and H_eval as we focused on because there is no H_find since the model is not trained at all prior to pruning. Furthermore, as per the extensive recent study in [9], pruning at initialization is not currently much better than random, and iterative magnitude pruning (where the process of determining the mask requires repeatedly training the model) is still significantly better, thus we focus on IMP. We have added a section to the related work discussing pruning at initialization and the differences to our work.
> - Within the lottery ticket framework, there is the original procedure where after pruning, weights are rewound back to their original initializations. Only in followup papers is late rewinding used. Both the original (rewind to iteration 0) and the late rewinding framework (rewind to iteration > 0) are valid methods; while late rewinding works better in some cases, the actual lottery ticket hypothesis (the claim that the original network contains a subnetwork that can do well on its own) requires rewinding to initialization, and ideally, LTs should work without late rewinding, which make the winning tickets data-dependent. Thus, we argue that it is valuable to study both late rewinding and rewinding to init.
>
> Similarly, “Since all of these experiments consider rewinding to an unstable phase of training (i.e. back to the random initialization or a very early in training), I am really uncertain whether these conclusions hold for LTs or pruning at initialization.” And for rewinding to a stable phase, “use a version of Algorithm 1 that rewinds to a stable training phase, c.f. [3, 8], where it has been previously shown that valid LTs can be found using standard IMP”:
> - In our late rewinding experiments (Figure 4a), we rewound to 2 epochs because it was around ~2% of total training, which is comparable to past papers. For instance, in “One ticket to win them all” (Morcos et al., NeurIPS 2019), the experiments used late rewinding to 1 epoch for Fashion MNIST, SVHN, CIFAR-10, and CIFAR-100 and 3 epochs for ImageNet and Places365.
> - However, we agree that analyzing the stability of the late rewinding epoch we chose is an interesting and important experiment. We have therefore added linear mode connectivity analyses in Figure A13, following the procedure in [3]. We also added Figure A14, which shows the winning ticket with late rewinding to 2 epochs performing significantly better than a baseline with the same mask but reinitiazlied weights. These results show that late rewinding to epoch 2 is indeed stable, verifying that our late rewinding results do apply to the stable phase of training.
>
> Why Tiny ImageNet:
> - As described by the reviewer, lottery tickets show different effects in different regimes, and many findings on small datasets do not generalize to larger datasets. Thus, we strove to go to as large a dataset as possible, but since IMP requires training a full model 30 times sequentially (plus separate trainings for each model evaluation when H_find /= H_eval), ImageNet is extremely costly and is not practical for the large suite of experiments we performed, so we focused on Tiny Imagenet. However, as suggested, we are currently running the core experiments on ImageNet (which take 1-2 weeks to complete), and plan to include them in the paper as soon as they finish.
>
> “Why would I use IMP to find a pruning mask at initialization if there is more efficient methods [4-7] for that?”
> - The paper you also mentioned [9], which rigorously evaluated methods for pruning at initialization, shows that IMP is substantively better than pruning at initialization and that pruning at initialization often only slightly beats random masks. We therefore argue that there is still substantial value to studying IMP.
>
> “Please clarify in the introduction that all observations are entirely limited to magnitude pruning.”
> - Thank you for this suggestion. We agree that this could have been made clearer in the original manuscript. We have added clarifications to both the intro and methods as requested. We also hope that the title and methods are clear in stating that we use magnitude pruning, and we make no mentions to pruning besides magnitude pruning anywhere in the paper.

---

> > ### Comment · AnonReviewer2 · 2020-11-20
> > **Some additional comments**
> >
> > ### Overall comment
> >
> > Thank you for your response and clarifying some of my concerns. I also appreciate the additional experiments that underscore the main finding of the paper, which I believe can be summarized as follows:
> >
> > Performant pruning masks that can be used to train sparse NNs from a given initialization (or from early in training) and are found using IMP may require different hyperparameters during the mask-finding stage versus the sparse training phase.
> >
> > I have read all the reviews and all your responses, however, my overall assessment largely remains unchanged. Mostly, my main concern that the results are open to different conflicting interpretations remains or as, e.g., R3 put it in their original review it is _"hard to tell what the practical benefit of these findings are."_
> >
> > In particular:
> >
> > * The LTH framework is exciting to me and provides practical benefits because it shows that in theory we could just halt training at an early stage (or even at initialization but only under certain settings!) and continue training with the sparse network. So if we had an efficient "oracle" (other than IMP) that could provide the pruning mask we could effectively remove large portions of the network early on. Naturally, this is an exciting observations that could lead to practical algorithms that provide such an efficient oracle.
> >
> > * On the other hand, your work provides a different IMP-based oracle to find those masks. This observation, however, is limited in its practical benefits. What can I do with this observation? It seems that it does not pave a way towards more efficient oracles but rather just shows that IMP-based masks dependent on the hyperparameters. It is also not really useful for finding a sparse network that performs well if we don't worry about overall training time, right?  Because then we can just use iterative learning rate rewinding [8 from my initial review].
> >
> > ### Some detailed responses
> >
> > ___
> >
> > _"We emphasize that we do not investigate pruning at initialization at all."_
> >
> > Thank you for the clarification. I agree that you don't study pruning at initialization. However, my point was mostly related to the practical benefits of your experiments. Unlike LTH, the conclusions of your experiments may not be directly beneficial in designing more efficient pruning algorithm as mentioned above.
> >
> > ___
> >
> > _"Thus, we argue that it is valuable to study both late rewinding and rewinding to init."_
> >
> > I don't really agree with this statement. Rewinding to init only works in very specific scenarios. So as a practioner I most likely don't want to spend the time and resources required in order to understand whether my particular combination of data set and architecture enables me to find performant LTs at the time of initialization. Rewinding to a stable phase of training is thus much more practical and general setting.
> >
> > ___
> >
> > _"These results show that late rewinding to epoch 2 is indeed stable, verifying that our late rewinding results do apply to the stable phase of training."_
> >
> > Unfortunately, I have to disagree again. I looked at Figures A13, A14 and they actually just confirm what I was worried about. In particular, Figure A13 highlights that rewinding to epoch 2 is not stable across all your tested hyperparameters but rather that it is only stable for small learning rates.
> >
> > ___
> >
> > _"we are currently running the core experiments on ImageNet"_
> >
> > Thank you very much. I look forward to seeing those results.
> >
> > ___
> >
> > _"We therefore argue that there is still substantial value to studying IMP."_
> >
> > Could you please clarify what you mean with "substantial value"? I also read  the paragraph on "Practical implications" in your updated paper but I wasn't really able to grasp the take-away message.

---

### Official Review · AnonReviewer3 · 2020-10-28
**Official blind review #3**

**Rating:** 5
**Confidence:** 3

**Review:**

##########################################################################

Summary:

In the lottery ticket hypothesis's general framework, network pruning's two phases of training (before and after the network is actually pruned) use the same hyperparameters - in particular, the learning rate schedule is identical in both phases.  Using different learning rates for these two phases, and, unintuitively, preferring a learning rate in phase 1 (finding the mask) that results in a *worse* dense model, can result in a better final model (after training with the mask in place).  It is also shown that the layerwise pruning ratios may be the key to understanding this behavior: finding the proper LPR is best done with a small learning rate, but given an LPR, a large learning rate is better able to determine the specific mask.

##########################################################################

Reasons for score:

I rated this submission as marginally below the acceptance threshold mainly because I'm struggling to find the practical benefit of the findings.  On one hand, decoupling learning rates seems to work for the limited results contained in the submission, but this boils down to "here's an extra hyperparameter we should be testing."  (This hyperparameter is also not new in all contexts, only in the LTH.)  On the other hand, these results pointed to the LPR as being crucial for a good pruned model.  Back to the first hand, though, what do we do with this information?  The submission doesn't offer any guidance about where to go next, except that perhaps other hyperparameters should be similarly decoupled, which isn't the most comforting thought for practitioners.  Finally, the claim that decoupling learning rates is brand new should be tempered somewhat (see below).

##########################################################################

Pros:

+ This paper is well-written and fairly easy to follow with a straightforward organization.
+ Decoupling learning rates in the LTH framework seems novel and useful.
+ Not argued by the authors, but I'll add/clarify: while decoupling learning rates might not be new for all pruning methods (traditionally pruning+fine-tuning, as opposed to rewinding), it *is* novel to use a lower learning rate during the mask finding phase, which may result in a worse dense model, from which a better sparse model is created.  (I don't think this has been validated empirically anywhere for prune+fine-tune approaches, so maybe it's best left as future work and the claim re-worded to not run afoul of past work that uses different learning rates.)
+ The importance of a good LPR is clearly motivated, and the experiments seem sufficient to prove the causal relationship.

##########################################################################

Cons:

- It's hard to tell what the practical benefit of these findings are.  I'd have expected a new SOTA result for some task with the LTH framework and decoupled learning rates, or at least a clear comparison of "the best baseline LTH vs. the best decoupled LRs LTH over a range of sparsities," but I can't find such a result anywhere.
- The discussion suggests practitioners incorporate "simple additional sweeps of decoupled learning rates."  These sweeps may be simple (they may not, depending on the training framework), but they will almost certainly be very expensive for most production data sets.  (The cost is further compounded when considering the early pruning results of section 3.4.)  Useful guidance for limiting costly hyperparameter sweeps, either from a theoretical grounding or wider empirical studies, would make this requirement less painful.
- Decoupling learning rates before and after pruning in general is nothing new as claimed in the introduction; past work (e.g. Han et al., 2015, referenced in the submission) specifically use different learning rates before and after pruning, so novelty/scope takes a slight hit. That said, applying it to the LTH framework is new, and it uncovered the dependence of a good model on finding a good LPR set.  (Also, as I argued *for* the submission above, it's new that the goal of the mask finding stage isn't necessarily to find the best-performing dense model; a worse dense model can lead to a better pruned model.)

##########################################################################

Questions:

- In section 3.2's late rewinding experiments, is the LR warmed up for both the find and eval phases?
- At the end of 3.2's structured pruning: "... decoupling LRs does not just apply to lottery tickets, but rather applies to global magnitude pruning in general."  Isn't this experiment still essentially the LTH framework, but with structure?  Application to "magnitude pruning in general" is too a broad claim, given past work (again, Han et al., 2015).  Or, if there was some experimental setup other than train, prune (with structure), rewind, etc., that is decidedly *not*-LTH, please clarify.
- The text description of Figure 5 describes different pruning iterations -- what are these iterations?

##########################################################################

Minor suggestions:

- Please add the particular network/data set used for each figure or table in the captions.
- There's a double "linear" in the LR warmup description (just one is enough).
- Similarly, the first sentence of 3.3 has a double "also."

---

> ### Author Response · Authors · 2020-11-20
> **Thank you for the review; here is our response**
>
> **Response to concerns:**
>
> “I'd have expected a new SOTA result for some task with the LTH framework and decoupled learning rates, or at least a clear comparison of ‘the best baseline LTH vs. the best decoupled LRs LTH over a range of sparsities,’”
> - Thank you for the great suggestion; we admit that the original Figure 2 does not clearly show the advantage of decoupling. We added Figure 3 to directly compare the best decoupled run and best coupled run at each sparsity level. In all cases, the best decoupled run outperforms the best coupled run for all sparsities, with gains of 3-4% for sparsities below 98%. We also note that the improvement is noticeably larger for batch size and weight decay (Figure 3b, 3c).
>
> “Decoupling learning rates seems to work for the limited results contained in the submission, but this boils down to ‘here's an extra hyperparameter we should be testing,’” “what do we do anyway with this information?” and “Useful guidance for limiting costly hyperparameter sweeps, either from a theoretical grounding or wider empirical studies, would make this requirement less painful.”
> - While we agree hyperparameter sweeps can be expensive, we would argue that it is strictly better to be aware of the fact that you might need to tune something than not. In current practice for lottery tickets, this axis is never even examined. It can help debug poorly performing LTs, and help improve accuracy for those who want to optimize for accuracy and can afford to do some additional tweaking.
> - However, we agree that general guidance would be extremely useful. We have therefore added a new paragraph to the discussion section directly explaining practical guidance. Briefly, H_eval should be similar to H_unpruned at low sparsity levels, LR_eval should be dropped slightly for at higher sparsity levels, and H_find requires a smaller learning rate and weight decay, or a larger batch size. Further, we note that turning off weight decay during mask discovery consistently results in better LPR, and consequently, better performing masks. Finally, we observed that early pruning can result in masks which are both higher performance and can substantially reduce the compute efficiency of finding lottery tickets.
>
> “The claim that decoupling learning rates is brand new should be tempered somewhat”
> - We thank the reviewer for this comment and acknowledge that this claim should be tempered; we rephrased our writing to reflect that decoupling LRs is only new for lottery tickets. However, while it is not new for fine-tuning methods, we show that fine-tuning (at low learning rates) does not work well because LR_eval needs to be larger; while [8] compared fine-tuning to learning rate rewinding, they did not make the connection that LR_eval is why learning rate rewinding worked better than fine-tuning. Additionally, we added experiments with decoupling batch size and weight decay, which is novel to the best of our knowledge.
>
> **Response to questions:**
>
> “In section 3.2's late rewinding experiments, is the LR warmed up for both the find and eval phases?”
> - We do not use warmup for late rewinding experiments, but use them independently as late rewinding was partially proposed as a replacement for warmup (Frankle, et al., ICML 2020). For our warmup experiments, we use the same warmup (linear, 1000 iterations) for both find and eval.
>
> Application to "magnitude pruning in general" is too broad:
> - We have rephrased the scope of our claims. We used that phrase originally because structured pruning (as done in Liu et al 2017) is not the same as the lottery ticket framework: we do not rewind the weights back to initialization, and we do one-shot pruning rather than IMP. The only things in common between that and lottery ticket is that they use global pruning and magnitude pruning, thus we felt like that was the best term to encapsulate both sets of experiments. However, we do realize that there may be other forms of global magnitude pruning that are not captured.
>
> Different pruning iterations
> - Thank you for pointing that out! We have fixed the text and added some additional sparsity levels / pruning iterations to the appendix.

---

> > ### Comment · AnonReviewer3 · 2020-11-23
> > **Response follow-up**
> >
> > I thank the authors for their continued effort and thoughtful response.  The newly-added Figure 3 in particular makes it easy to compare LT results with and without the submission's proposed techniques. My overall rating still stands, however:
> >
> > The main conclusion of the submission is that decoupling hyperparameters between the mask-finding and mask-evaluation phases of the Lottery Ticket Hypothesis can result in models with higher accuracy for a given level of sparsity.  In hindsight, this isn't surprising, as it is often the case in other training/pruning frameworks.  That obviousness in hindsight would be forgivable if there were something immediate practitioners could use (just because something is now obvious doesn't make it not novel or useful).  Lacking real suggestions (the new heuristics are a bit scattered - do I want to disable Weight Decay or not?), we're left with a mostly novel, but unsurprising, result.
> >
> > Sure, practitioners interested in accuracy at all costs could find it worthwhile to sweep all these hyperparameters within the LTH framework.  However, other methods for inducing sparsity could be simpler _and_ more accurate; without comparisons to other SOTA methods, the findings within this submission aren't sufficient to raise my rating.
> >
> > -----
> > Minor typo in the new version: in the limitations section, there's an instance of "LRP" that should be "LPR."

---

### Author Response · Authors · 2020-11-20
**Notes on updated paper**

We thank all the reviewers for their detailed and insightful feedback. We were happy to see that the reviewers found our paper to be “very clean and well-written” (R2) with “really interesting” results (R2) and that “the analysis is also strong” (R1). We were also glad that they think that our ideas are “novel and useful” (R3), “an extremely important research direction in network pruning” (R4) with “the potential to change how people think about network pruning” (R1).

We will address each reviewer’s concerns as individual response comments. As a general comment, since initial submission, we have broadened the scope of our paper to include additional hyperparameters: in addition to learning rate (LR), we ran the same experiments on batch size (BS) and weight decay (WD) as well. For the main experiments, we adjusted LR, BS, and WD independently. Across hyperparameters, our core findings for LR generally hold true: the BS and WD used to find the masks should be different from those used to evaluate masks as better performing BS/WD can yield worse masks. Further, this effect is causally mediated by the layerwise pruning ratios (LPR) for both BS and WD. Additionally, we tuned LR and WD together and found that when WD is set to zero, the effect of LR_find is changed. We hope that these new results increase the generality, novelty, and impact of our contributions. Please see our individual responses to reviewers and the updated paper for details of these experiments.

---

### Decision · Program_Chairs · 2021-01-07
**Final Decision**

**Decision:**

Reject

**Comment:**

This paper explores the role of hyperparameters in the separate phases of a classic pruning pipeline: mask identification and retraining. Key observations include a set of the hyperparameters to search relative to a standard regime as well as the identification that the layerwise pruning rates from mask finding are intertwined with these hyperparameters and are what chiefly affects the eventual performance of the pruned network.

The pros of this paper are that it works against the contemporary wisdom that the default hyperparameters for a model are the best for finding a mask for the model. Instead, there are improvements to be had by identifying a set of hyperparameters that lead to worse overall model accuracy, but better masks. Second, the work shows that the layerwise pruning rates are the key elements of these hyperparameters effect. The rates can in fact be transferred to more poorly performing network configurations and improve performance.

The cons of this paper, as noted by the reviewers, are the somewhat unclear implications of the technique. The added guidance on directions to improve hyperparameters is valuable but does not necessarily provide a cost-effect strategy to find these. At its strongest, this guidance offers practitioners a recommendation to also consider hyperparameters for the initial model.

The stronger, forward-looking implication is, instead, the connection to layerwise pruning rates. Specifically, while layerwise pruning rates have been demonstrated to be important in the literature (e.g., [1]), there has been a limited study into the exact nature of a good set of pruning rates versus a bad set of pruning rates.  Where this paper stops short of a clear result, is if were to connect excessive pruning of the earlier layers, or simply the layerwise rates themselves to another property of the network (e.g., gradient flow, or capacity) that indicates the improved eventual performance.

My Recommendation is to Reject. The paper's core experiments are well-executed. However, this final detail, closing the gap between the portability of these layerwise rates and a conceptual understanding, is a key missing component.  Once done, that will make for a very strong paper.

[1] AMC: AutoML for Model Compression and Acceleration on Mobile Devices. Yihui He, Ji Lin, Zhijian Liu, Hanrui Wang, Li-Jia Li, Song Han. EECV, 2018